# MultiPDENet: PDE-embedded Learning with Multi-time-stepping for Accelerated Flow Simulation

**Qi Wang** [1]   **Yuan Mi** [1]   **Haoyun Wang** [1]   **Yi Zhang** [2]   **Ruizhi Chengze** [2]
**Hongsheng Liu** [2]   **Ji-Rong Wen** [1]   **Hao Sun** [1]

## Abstract

Solving partial differential equations (PDEs) by numerical methods meet computational cost challenge for getting the accurate solution since fine grids and small time steps are required. Machine learning can accelerate this process, but struggle with weak generalizability, interpretability, and data dependency, as well as suffer in long-term prediction. To this end, we propose a PDE-embedded network with multiscale time stepping (MultiPDENet), which fuses the scheme of numerical methods and machine learning, for accelerated simulation of flows. In particular, we design a convolutional filter based on the structure of finite difference stencils with a small number of parameters to optimize, which estimates the equivalent form of spatial derivative on a coarse grid to minimize the equation's residual. A Physics Block with a 4th-order Runge-Kutta integrator at the fine time scale is established that embeds the structure of PDEs to guide the prediction. To alleviate the curse of temporal error accumulation in long-term prediction, we introduce a multiscale time integration approach, where a neural network is used to correct the prediction error at a coarse time scale. Experiments across various PDE systems, including the Navier-Stokes equations, demonstrate that MultiPDENet can accurately predict long-term spatiotemporal dynamics, even given small and incomplete training data, e.g., spatiotemporally down-sampled datasets. MultiPDENet achieves the state-of-the-art performance compared with other neural baseline models, also with clear speedup compared to classical numerical methods.

[1]Gaoling School of Artificial Intelligence, Renmin University of China, Beijing, China [2]Huawei Technologies, Shenzhen, China. Correspondence to: Hao Sun <haosun@ruc.edu.cn>.

*Proceedings of the $42^{nd}$ International Conference on Machine Learning*, Vancouver, Canada. PMLR 267, 2025. Copyright 2025 by the author(s).

## 1. Introduction

Complex spatiotemporal dynamical systems, e.g., climate system (Schneider et al., 2017) and fluid dynamics (Ferziger et al., 2019), are fundamentally governed by partial differential equations (PDEs). To capture the intricate behaviors of these systems, various numerical methods have been developed. Direct Numerical Simulation (DNS) is a widely used method for solving PDEs. It requires specifying initial conditions (ICs), boundary conditions (BCs), and PDE parameters, followed by discretizing the equations on a grid using techniques like finite difference (FD), finite element (FE), finite volume (FV), or spectral methods. Despite their accuracy, traditional numerical methods face key challenges of high computational costs (Goc et al., 2021), when addressing with high-dimensional problems or necessitating fine spatial and temporal resolutions.

Recent advances in deep learning have introduced neural-based approaches (Lu et al., 2021; Li et al., 2021; Gupta & Brandstetter, 2023) for solving PDEs. These data-driven methods eliminate the need for explicit theoretical formulations, enabling networks to learn underlying patterns directly from data through end-to-end training. While promising, these approaches face notable challenges, including a heavy dependence on large training datasets and limited generalization. For instance, achieving accurate predictions becomes particularly challenging when models encounter unseen ICs or scenarios beyond the training distribution.

A notable progress lies in the paradigm of Physics-informed neural networks (PINNs) (Raissi et al., 2019), which incorporates physical prior knowledge (such as PDE residuals and I/BCs) as constraints within the loss function. This approach allows the network to fit the data while simultaneously maintaining a certain degree of physical consistency. Variants of PINNs (Raissi et al., 2020; Wang et al., 2020; Eshkofti & Hosseini, 2023) have shown notable success across various domains, reducing the dependency on extensive datasets to some degree. However, such methods still face scalability and generalizability challenges when applied to complex nonlinear dynamical systems. Additionally, optimizing complex loss functions (Rathore et al., 2024) and ensuring model interpretability remain challenges.

A series of approaches have been proposed to integrate physics into neural networks (NNs) to overcome the above challenges. For instance, PeRCNN (Rao et al., 2022; 2023), which uses feature map multiplication to construct polynomial combinations for approximating the underlying PDEs, can capture the latent spatiotemporal dynamics even with low-resolution, noisy, and coarse data, demonstrating strong generalizability. Nevertheless, this method suffers from error accumulation, degrading its performance in long-term predictions. Another approach (Kochkov et al., 2021; Sun et al., 2023), combining NNs with numerical methods, aims to accelerate the simulation process on coarse grids. These hybrid methods leverage traditional solvers for stability and NNs for accuracy. However, they often rely heavily on NN capabilities and often requires large amounts of data.

To overcome these limitations, we propose MultiPDENet, a PDE-embedded network that incorporates multiscale time-stepping (as shown in Figure 1), to efficiently simulate spatiotemporal dynamics, e.g., turbulent fluid flows, on coarse spatial and temporal grids with limited data. Notably, it integrates a trainable neural solver for precise predictions at micro time scales, while employing a NN to correct errors at macro time steps. Additionally, by embedding PDEs, MultiPDENet offers enhanced generalizability. The primary contributions of this work are summarized as follows:

- We developed MultiPDENet, a PDE-embedded network with multiscale time-stepping, for accelerated flow simulations on spatiotemporal coarse grids. By integrating neural solver with PDEs, MultiPDENet achieves great generalizability and efficiency.

- Leveraging the structure of FD stencils, we introduced a symmetric convolutional filter that approximates the equivalent form of derivatives on coarse grids, aiming to reduce the residual error of the governing PDEs.

- Experimental results across various datasets, covering 1D and 2D equations (e.g., reaction-diffusion processes and turbulent flows), demonstrate the effectiveness of MultiPDENet in accelerating long-term simulations.

## 2. Related Work

Related works on numerical, machine learning, physics-inspired learning, and hybrid learning methods for simulation of PDE systems are given in Appendix A.

## 3. Methodology

### 3.1. Problem Description

Let's consider a general spatiotemporal dynamical system governed by the following PDE:

$$\mathbf{u}_t = \mathcal{F}(\mathbf{u}, \mathbf{u}^2, \ldots, \boldsymbol{\nabla}\mathbf{u}, \Delta\mathbf{u}, \ldots; \boldsymbol{\lambda}) + \mathbf{f}, \quad (1)$$

where $\mathbf{u}(\mathbf{x}, t) \in \mathbb{R}^n$ denotes the physical state in the spatiotemporal domain $\Omega \times [0, T]$; $\mathbf{u}_t$ the first-order time derivative term; $\mathcal{F}(\cdot)$ a linear/nonlinear functional parameterized by PDE parameters $\boldsymbol{\lambda}$ (e.g., the Reynolds number $Re$); $\boldsymbol{\nabla}$ the Nabla operator is defined as $[\partial x, \partial y, ...]^{\mathrm{T}}$; and $\mathbf{f}$ the source term. Additionally, we define $\mathcal{I}(\mathbf{u}, \mathbf{u}_t; \mathbf{x} \in \Omega, t = 0) = \mathbf{0}$ and $\mathsf{B}(\mathbf{u}, \boldsymbol{\nabla}\mathbf{u}, \cdots; \mathbf{x} \in \partial\Omega) = \mathbf{0}$ specified ICs and BCs, where $\partial\Omega$ represents the domain boundary.

We aim to accelerate the simulation of fluid flows by using a PDE-embedded network with multiscale time stepping based on a limited training data (coarse in both spatial and temporal scales). The model is capable of rapid simulation, achieving high solution accuracy while demonstrating strong generalizability across varying ICs, source terms, complex domains, and PDE parameters.

### 3.2. Model Architecture

In this section, we introduce MultiPDENet and show how our model efficiently captures the underlying spatiotemporal dynamics. As illustrated in Figure 1(**a**), predicting $\mathbf{u}_{k+1}$ from the input $\mathbf{u}_k$ involves two main components: the Physics Block and the $\mathrm{M_aNN}$ Block.

#### 3.2.1. MULTI-SCALE FORWARD TIME STEPPING SCHEME

While the learnable neural solver can be used independently, its accuracy for long-term prediction is limited due to error accumulation. To address this issue, we introduce a multi-scale time stepping scheme, incorporating micro-scale and macro-scale steps, to improve predictive accuracy and enables fast prediction of PDE solutions on coarse spatiotemporal grids. Specifically, we define two types of time stepping: micro-scale step and macro-scale step, to enhance the performance of spatiotemporal dynamics prediction. At the macro scale, given the coarse solution $\mathbf{u}_k$ at time $t_k$, MultiPDENet is expected to predict the next-step solution $\mathbf{u}_{k+1}$ at $t_{k+1}$, which can be expressed as:

$$\mathbf{u}_{k+1} = \mathbf{u}_k + \sum_{m=1}^{M} \delta\bar{\mathbf{u}}_m^k + \mathrm{M_aNN}(\mathbf{u}_k, \Delta t, dx), \quad (2)$$

where $\Delta t$ is the macro-scale time interval, and $dx$ the spatial resolution of mesh grid. Here, $\delta\bar{\mathbf{u}}_m^k$ is the incremental update by the Physics Block (see Section 3.2.2) at each micro step, as shown in Eq. (3), where $M$ denotes the number of micro-scale time steps in one macro-scale step (e.g., $M = 4$ herein). The $\mathrm{M_aNN}$ Block (see Section 3.2.4) refines these incremental updates generated by the Physics Block on coarse grids, yielding the final update for the macro step.

#### 3.2.2. PHYSICS BLOCK: LEARNABLE NEURAL SOLVER

To accurately predict at the micro-scale step, we developed a neural solver, referred to as the Physics Block, as illustrated

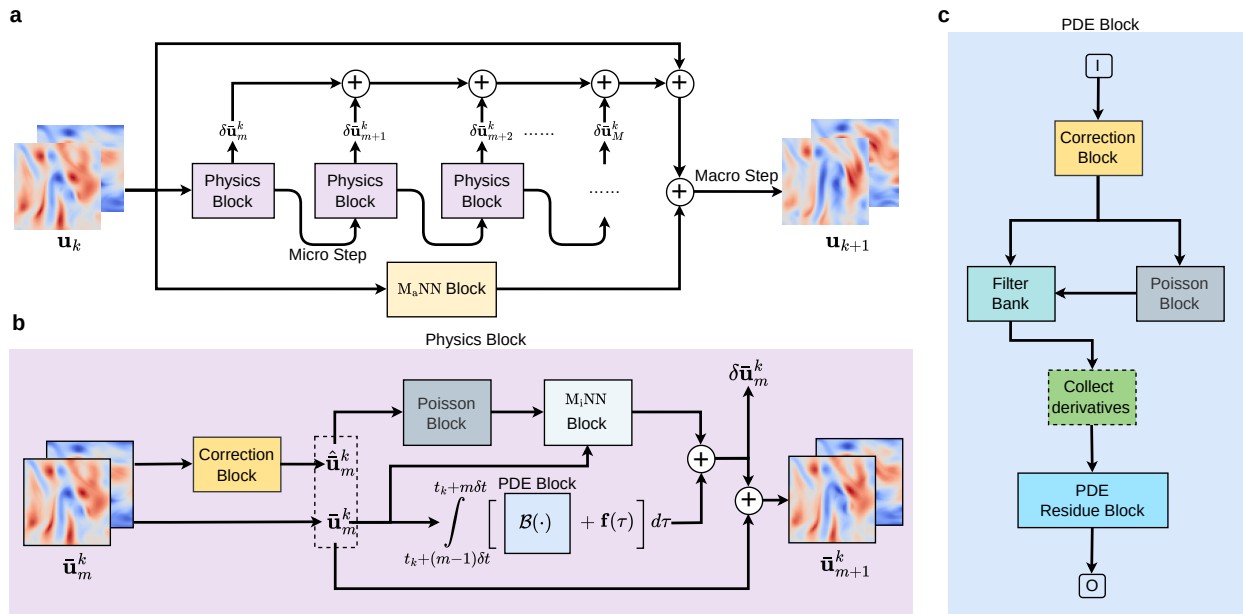

*Figure 1.* Schematic of MultiPDENet for learning turbulent flows. (**a**), Model architecture. (**b**), Physics Block. (**c**), Learnable PDE block.

in Figure 1(**b**). This solver is designed to ensure the stability (Hoffman & Frankel, 2018), accuracy, and efficiency of its predictions by adhering to the Courant-Friedrichs-Lewy (CFL) conditions (LeVeque, 2007). The Physics Block comprises three main components: the Poisson Block, the PDE Block, and the $M_iNN$ Block. The solution update for each micro-scale time step can be describe as $\bar{\mathbf{u}}_{m+1}^k = \bar{\mathbf{u}}_m^k + \delta\bar{\mathbf{u}}_m^k$, where

$$\delta\bar{\mathbf{u}}_m^k = \int_{t_k+(m-1)\delta t}^{t_k+m\delta t} \left[ \mathcal{B}\left(\tilde{\mathbf{u}}(\tau), \boldsymbol{\nabla}\tilde{\mathbf{u}}(\tau), \cdots; \boldsymbol{\lambda}\right) + \mathbf{f}(\tau) \right] d\tau$$
$$+ M_iNN\left( \bar{\mathbf{u}}_m^k, \Xi_m^k\left(p, \hat{\boldsymbol{\nabla}}\hat{\mathbf{u}}, \hat{\boldsymbol{\nabla}}^2\hat{\mathbf{u}}, \hat{\boldsymbol{\nabla}}p, \mathbf{f}, Re\right) \right). \quad (3)$$

Here, $\bar{\mathbf{u}}_m^k$ represents the intermediate state at $m$-th micro-scale step initialized at time $t_k$ (note that $\bar{\mathbf{u}}_1^k = \mathbf{u}_k$). We denote $\tilde{\mathbf{u}}(\tau) \triangleq \mathbf{u}(\tilde{\mathbf{x}}, \tau)$, where $\tilde{\mathbf{x}}$ depicts the coordinates of coarse grid. Moreover, $\boldsymbol{\lambda}$ can be set as trainable if unknown. $\mathcal{B}$ represents the PDE block, used for approximating $\mathcal{F}$ in Eq. (1). To keep the accuracy and ensure the stability, the PDE block is designed based on the RK4 integrator (see Appendix B.3) and consists of the Correction Block and a trainable filter bank. Since the considered micro-scale time interval is relatively large, the $M_iNN$ Block is used as a corrector to refine the solution. More details can be found in Appendix B. In fact, the Physics Block can be used for prediction independently (e.g., the quantitative results for the NSE dataset predictions using purely the Physics Block are presented in Table 3, labeled Model C).

**PDE Block.** The PDE block computes the residual of the governing PDEs. It incorporates a learnable filter bank with

symmetry constraints, which calculates derivative terms based on the corrected solution produced by a Correction Block. These terms are then combined into the governing PDEs, a learnable form of $\mathcal{F}$ in Eq. (1). This process is incorporated into the RK4 integrator (see Appendix B.3) for solution update which can be expressed as

$$\mathcal{F}\left( \bar{\mathbf{u}}_m^k, \cdots, \hat{\boldsymbol{\nabla}}\hat{\mathbf{u}}_m^k, \hat{\boldsymbol{\nabla}}^2\hat{\mathbf{u}}_m^k, \cdots; \boldsymbol{\lambda} \right)$$
$$\xleftarrow{approx.} \mathcal{B}\left( \bar{\mathbf{u}}_m^k, \cdots, \boldsymbol{\nabla}\bar{\mathbf{u}}_m^k, \boldsymbol{\nabla}^2\bar{\mathbf{u}}_m^k, \cdots; \boldsymbol{\lambda} \right), \quad (4)$$

where $\mathcal{B}$ denotes the PDE block, and $\bar{\mathbf{u}}_m^k$ the coarse solution (aka, solution on coarse grids) at micro-scale time $t_k + m\delta t$. Here, $\hat{\mathbf{u}}_m^k$ refers to the neural-corrected state of the coarse solution, which is obtained through the Correction Block (see Appendix B.1 for details). This corrected state $\hat{\mathbf{u}}_m^k$ is used to estimate spatial derivatives, namely, $\hat{\mathbf{u}}_m^k = NN(\bar{\mathbf{u}}_m^k)$. Note that $\hat{\boldsymbol{\nabla}}$ and $\hat{\boldsymbol{\nabla}}^2$ represent trainable Nabla and Laplace operators, respectively, each consisting of a symmetrically constrained convolution filter, e.g., an enhanced FD kernel to approximate spatial equivalent derivatives. By utilizing the RK4 integrator, we can project the coarse solution to the subsequent micro-scale time step. Despite the reduced resolution causing some information loss, this learnable PDE block enables a closer approximation of the equivalent form of the derivatives on coarse grids. This addition serves as a fully interpretable "white box" element within the overall network structure.

**Poisson Block.** In solving incompressible NSE, the pressure term, $p$, is obtained by solving an associated Poisson equation. To compute the pressure field, we imple-

*Table 1.* Overview of datasets and training configurations. Note that "→" denotes the downsampling process from the high resolution (simulation) to the low resolution (training and testing).

| Dataset | Numerical Method | Spatial Grid | Time Steps (Temporal Grid) | # of Training Trajectories | # of Testing Trajectories | Macro-step Rollout | Micro-step Rollout |
|---------|------------------|--------------|----------------------------|----------------------------|---------------------------|--------------------|--------------------|
| KdV | Spectral | $256 \rightarrow 64$ | $10000 \rightarrow 2000$ | 3 | 10 | 10 | 4 |
| Burgers | FD | $100^2 \rightarrow 25^2$ | $2000 \rightarrow 200$ | 5 | 10 | 10 | 4 |
| GS | FD | $128^2 \rightarrow 32^2$ | $4000 \rightarrow 200$ | 3 | 10 | 1 | 4 |
| NSE | FV | $2048^2 \rightarrow 64^2$ | $153600 \rightarrow 1200$ | 5 | 10 | 1 | 4 |

mented a pressure-solving module shown in Figure S1(a), which solves the Poisson equation, $\Delta p = \psi(\mathbf{u})$, where $\psi(\mathbf{u}) = 2\left(u_x v_y - u_y v_x\right)$ for 2D problems (the subscripts indicate the spatial derivatives along $x$ or $y$ directions). To compute the pressure, we employ a spectral method (Poisson solver) based on $\psi(\bar{\mathbf{u}}_m^k)$ to calculate $\bar{p}_m^k$. As shown in Figure S1(b), this approach dynamically estimates the pressure field from the velocity inputs, removing the need for labeled pressure data.

### 3.2.3. ADAPTIVE FILTER WITH CONSTRAINT

Traditional FD methods often yield inaccurate derivatives on coarse grids. To address this, we propose a learnable filter with constraints that approximates equivalent derivatives on coarse grids, minimizing the PDE residuals during training and thereby improving the model's predictive accuracy. By leveraging the symmetry of central difference stencils, our filter maintains structural integrity while enhancing network flexibility. As shown in Figure 2, we construct two $5 \times 5$ symmetric matrices, each requiring only six learnable parameters due to symmetry constraints. These matrices are designed to compute the first-order ($g'$) and second-order ($g''$) derivatives, respectively. In the matrix of $g''$, $s = 4 \times (a_3 + a_4 + a_5 + a_6) + 2 \times (a_1 + a_2)$. This design leverages the structural properties of central difference methods. By satisfying the Order of Sum Rules (Long et al., 2018), this filter can achieve up to 4th-order accuracy in approximating the derivatives via optimization of trainable parameters.

### 3.2.4. NN BLOCK

To alleviate the error accumulation during long-term predictions on coarse grids, we introduce the $\mathrm{M_iNN}$ and $\mathrm{M_aNN}$ Blocks, operating at micro- and macro-scales, respectively. The MiNN Block employs a lightweight model (e.g., FNO, DenseCNN (Liu et al., 2024a)) for efficient micro-step predictions, whereas the $\mathrm{M_aNN}$ Block delivers more accurate predictions at larger steps (Gupta & Brandstetter, 2023). In this study, we utilized FNO as the $\mathrm{M_iNN}$ Block and UNet as the $\mathrm{M_aNN}$ Block. The significance of these blocks is evident from the ablation studies presented in Table 3.

**$\mathrm{M_iNN}$ Block.** The $\mathrm{M_iNN}$ block is designed to rectify error accumulation during micro-scale time step predictions. As

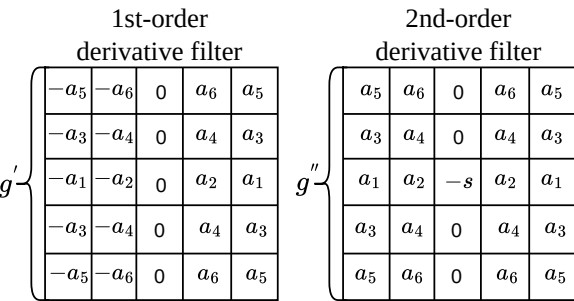

1st-order derivative filter — 2nd-order derivative filter

*Figure 2.* Symmetric filter

shown in Figure 1(b) in the upper path, $\bar{\mathbf{u}}_m^k$ is first corrected by the Correction Block, and $\bar{p}_m^k$ is computed by the Poisson Block. Inputs, including solution states $\{\bar{\mathbf{u}}_m^k, \bar{p}_m^k\}$ and their derivative terms, forcing term, and Reynolds number, are fed into the $\mathrm{M_iNN}$ Block (see Figure S1(d)). The $\mathrm{M_iNN}$ Block continuously refines the PDE block's outputs on the fly. For detailed information of the $\mathrm{M_iNN}$ Block settings, please refer to Appendix Table S5.

**$\mathrm{M_aNN}$ Block.** Although the Physics Block offers real-time corrections for the $\mathrm{M_iNN}$ outputs, errors still accumulate in long-term predictions. To mitigate error accumulation in long-term predictions given training data sampled at large time steps (e.g., $128\Delta t$ for the NSE dataset), we introduce the $\mathrm{M_aNN}$ Block. As depicted in Figure 1(a), the $\mathrm{M_aNN}$ Block takes the current velocity field $\mathbf{u}_k$ as input, and updates the solution $\mathbf{u}_{k+1}$ which is obtained by integrating the outputs from both the upper and lower paths. During the backpropagation, the $\mathrm{M_aNN}$ Block learns to correct the coarse solution output of the Physics Block in real time, ensuring that their combined results more closely align with the ground truth. The configuration details for this block are found in Appendix Table S6.

## 4. Experiment

We validate the performance of our method against baseline models on various PDE datasets. We then perform generalization tests across different external forces ($\mathbf{f}$), Reynolds numbers ($Re$), and domain sizes on the Kolmogorov flow (KF) dataset. Finally, we present ablation studies to demonstrate the contributions of each component in our model.

*Table 2.* Results of MultiPDENet and baselines. For KdV, Burgers, and GS, we inferred upper time limits of 50 s, 1.4 s, and 1200 s, for the test set as the system dynamics stabilized within these trajectories. These time limits were used to calculate HCT.

| Case | Model | RMSE ($\downarrow$) | MAE ($\downarrow$) | MNAD ($\downarrow$) | HCT (s) |
|---|---|---|---|---|---|
| KdV | FNO | 0.9541 | 0.4607 | 0.3469 | 10.0833 |
| | PINO | 0.4120 | 0.3022 | 0.2139 | 13.90 |
| | UNet | 1.9887 | 1.5722 | 1.6158 | 3.1250 |
| | DeepONet | NaN | NaN | NaN | 0.1500 |
| | **MultiPDENet** | **0.1536** | **0.1110** | **0.0833** | **39.8** |
| | Improvement ($\uparrow$) | 62.7% | 63.3% | 61.1% | 186.3% |
| Burgers | FNO | 0.0980 | 0.0762 | 0.0620 | 0.3000 |
| | PINO | 0.0832 | 0.0749 | 0.0599 | 0.5546 |
| | UNet | 0.3316 | 0.2942 | 0.2556 | 0.0990 |
| | DeepONet | 0.2522 | 0.2106 | 0.1692 | 0.0020 |
| | PeRCNN | 0.0967 | 0.1828 | 0.1875 | 0.4492 |
| | **MultiPDENet** | **0.0057** | **0.0037** | **0.0031** | **1.4000** |
| | Improvement ($\uparrow$) | 93.1% | 95.1% | 94.8% | 152.4% |
| GS | FNO | 8774 | 1303 | 1303 | 270 |
| | PINO | 0.5721 | 0.3579 | 0.3520 | 510 |
| | UNet | NaN | NaN | NaN | 20 |
| | DeepONet | 0.4113 | 0.2961 | 0.2898 | 568 |
| | PeRCNN | 0.1763 | 0.1198 | 0.1198 | 640 |
| | **MultiPDENet** | **0.0573** | **0.0294** | **0.0298** | **1400.0** |
| | Improvement ($\uparrow$) | 67.5% | 75.5% | 75.1% | 118.8% |
| NSE | FNO | 1.0100 | 0.7319 | 0.0887 | 2.5749 |
| | UNet | 0.8224 | 0.5209 | 0.0627 | 3.9627 |
| | LI | NaN | NaN | NaN | 3.5000 |
| | TSM | NaN | NaN | NaN | 3.7531 |
| | DeepONet | 2.1849 | 1.0227 | 0.1074 | 0.1126 |
| | **MultiPDENet** | **0.1379** | **0.0648** | **0.0077** | **8.3566** |
| | Improvement ($\uparrow$) | 83.2% | 87.6% | 87.7% | 110.9% |

## 4.1. Setup

**Dataset.** We generate the data using high-order FD/FV methods with high resolution under periodic boundary conditions and then downsample it spatially and temporally to a coarse grid. The low-resolution dataset is used for both training and testing. We consider four distinct dynamical systems: Korteweg-de Vries (KdV), Burgers, Gray-Scott (GS), and Navier-Stokes equations (NSE). Each dataset is divided into 90% for training and 10% for validation. We segment trajectories into data series, where each sample includes multi snapshots (e.g., for the KdV dataset, the sample length is set to 10, as detailed in Table 1) separated by a time step $\Delta t$, the 2nd to the last snapshot serves as the training labels. During training, we use only 3–5 trajectories for each system, and evaluate them on 10 distinct trajectories. For further details, please refer to Appendix C.

**Model training.** Our objective is to accelerate flow simulations with all computations anchored to coarse grids. During training, the model solely predicts the solutions for subsequent time steps, employing Mean Squared Error (MSE) as the loss metric. Unlike PINNs, our MultiPDENet directly embeds PDEs into its architecture, resulting in a loss function that exclusively comprises data loss, given by: $\mathcal{J}(\boldsymbol{\lambda}) = \frac{1}{BN} \sum_{i=1}^{B} \sum_{j=1}^{N} MSE\left(\check{\mathbf{H}}_{ij}, \mathbf{H}_{ij}\right)$, where $\check{\mathbf{H}}_{ij}$ denotes the coarse solution predicted by model rollout for the $j$-th sample in the $i$-th batch, and $\mathbf{H}_{ij}$ is the correspond-

ing ground truth. Here, $N$ denotes the number of batches, $B$ the batch size, and $\boldsymbol{\lambda}$ the trainable PDE parameters.

**Model generalization.** The generalization of MultiPDENet is evaluated across ICs, PDE parameters (e.g., $Re$), force terms, and computational domain sizes (e.g., different mesh grids). The model integrates ICs through its time-marching mechanism, ensuring robust generalization when trained effectively. The Reynolds number ($Re$) is represented via a two-dimensional embedding, $Re_{\text{embb}} = \frac{1}{Re} \cdot (\mathbf{a} \otimes \mathbf{b})$, using trainable vectors $\mathbf{a}$ and $\mathbf{b}$. This embedding, applied in both the PDE and MiNN blocks, reduces error propagation from the diffusion term on coarse grids and enhances generalization across $Re$ values. The force term is incorporated into the learnable PDE block and the MiNN block, where it serves as both a PDE feature and an input feature map as shown in Figure S1, enabling joint learning of force variations for better generalization.

**Evaluation metrics.** We evaluate the performance of our model using four metrics: Mean Absolute Error (MAE), Root Mean Square Error (RMSE), Mean Normalized Absolute Difference (MNAD), and High Correlation Time (HCT). For detailed definitions, please refer to Appendix E.

**Baseline models.** To ensure a comprehensive comparison, we selected several baseline models, including FNO (Li et al., 2021), PINO (Li et al., 2024c), UNet (Gupta & Brandstetter, 2023), TSM (Sun et al., 2023), LI (Kochkov et al., 2021), DeepONet (Lu et al., 2021), and PeRCNN (Rao et al., 2023). Details are found in Appendix F.

## 4.2. Solving PDE Systems

**KdV.** The primary challenge of this dataset lies in accurately capturing the complex interplay between nonlinearity and dispersion, leading to phenomena like soliton formation (Gardner et al., 1967). As shown in Figure 3(**a**), each baseline model struggles to produce accurate predictions, with DeepONet exhibiting significant divergence. In contrast, MultiPDENet demonstrates superior accurate predictions for ICs outside the training range. The correlation curve in Figure 3(**b**) highlights the significantly higher correlation of MultiPDENet compared to the baselines. The error distribution in Figure 3(**c**) confirms its lower error levels. Table 2 shows our model's generalizability, with performance improvements ranging from 61.1% to 186.3%.

**Burgers.** As shown in Figure 3(**d**), the solution snapshots predicted by MultiPDENet are significantly more accurate than those of the baseline models. The baseline models, limited by the coarse training data, produce incorrect predictions. The correlation curve in Figure 3(**e**) shows that MultiPDENet maintains a high correlation with the ground truth throughout the prediction, while other baseline models diverge. This is further evidenced by the error distribution in Figure 3(**f**), demonstrating that MultiPDENet's error is

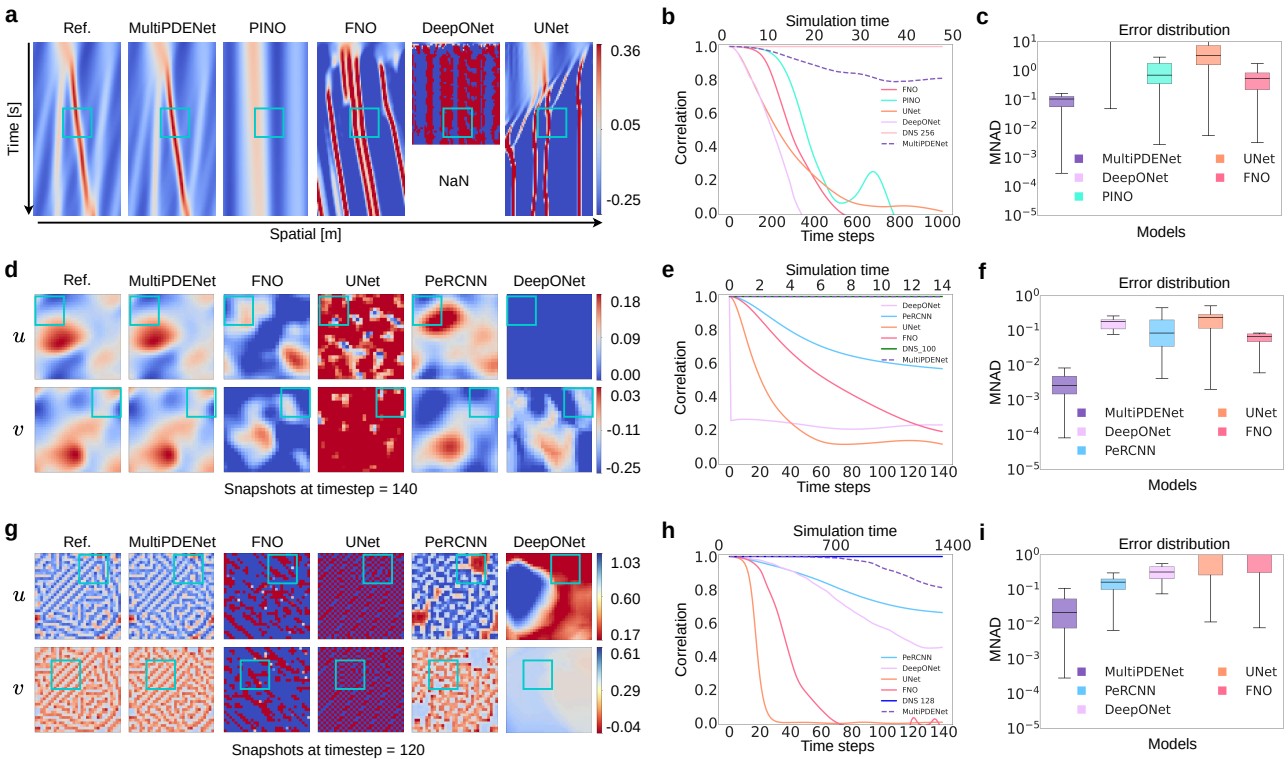

*Figure 3.* An overview of the comparison between our MultiPDENet and baselines, including predicted solutions (left), correlation curve (middle), and error distributions (right). (a)-(c) show the qualitative results on KdV. (d)-(f) show the qualitative results on Burgers. (g)-(i) show the qualitative results on GS. These PDE systems are trained with grid sizes of 64, 25×25, 32×32, respectively.

over an order of magnitude lower than that of the baselines. Table 2 confirms these findings with MultiPDENet's improvements exceeding 94.1% across all evaluation metrics.

**GS.** This reaction-diffusion system is nonlinear, making it challenging to capture its complex patterns (see Figure 3(**g**)). Only MultiPDENet accurately predicts the trajectory evolution. The baseline models struggle to learn the spatiotemporal dynamics, and even PeRCNN, despite its embedded physics, produces inaccurate predictions due to the limited and coarse training data. Figure 3(**h**) demonstrates the superior correlation of MultiPDENet's predictions with the ground truth. The error analysis in Figure 3(**i**) reveals significantly lower error levels for MultiPDENet, often by 1 to 2 orders of magnitude smaller compared to the baselines. Table 2 further validates this observation, with MultiPDENet's improvements of 67.5% to 118.8% over the best baseline.

**NSE**. We evaluate a KF with $Re = 1000$ across different ICs, governed by the NSE. Figure 4(**a**) shows the trajectory snapshots predicted by MultiPDENet and the baseline models over 10 s. Our model outperforms DNS 512, accurately capturing both global and local correction patterns. The neural methods, particularly FNO, exhibit poor generalization, producing granular and erroneous solutions. Among the Physics + ML baselines, TSM performs the best, but

starts to produce incorrect patterns at $t = 5$ s due to error accumulation. The correlation curve in Figure 4(**b**) supports these findings. Our model also achieves a spectrum energy distribution closely matching the ground truth (see Figure 4(**c**)). Table 2 highlights a performance improvement of over 83.2%. Even with 20% less training data, our model maintains strong generalizability (see Appendix Table S3).

### 4.3. Model Generalization

We conducted generalization tests on the KF flow dataset to assess our model's ability to capture the underlying dynamics. The model was initially trained on 5 sets of trajectories, where the forcing term is defined as $\mathbf{f} = \sin(4y)\boldsymbol{\eta}_x - 0.1\mathbf{u}$ with $Re = 1000$ and $\boldsymbol{\eta}_x = [1, 0]^T$. After training, we tested MultiPDENet on 10 different sets of trajectories, each with varying Reynolds numbers ($Re$) and forcing terms ($\mathbf{f}$), to evaluate its performance across a range of different ICs.

**Test on various Reynolds numbers.** Firstly, we evaluate the generalizability of MultiPDENet across four different Reynolds numbers: $Re = 500, 800, 1600, 2000$. The varying $Re$ values result in trajectories with differing levels of complexity. Figure 5(**a**) displays the accurate predictions made by our model at time step 300 for different $Re$. Figure 5(**b**) shows the correlation curves between the predicted

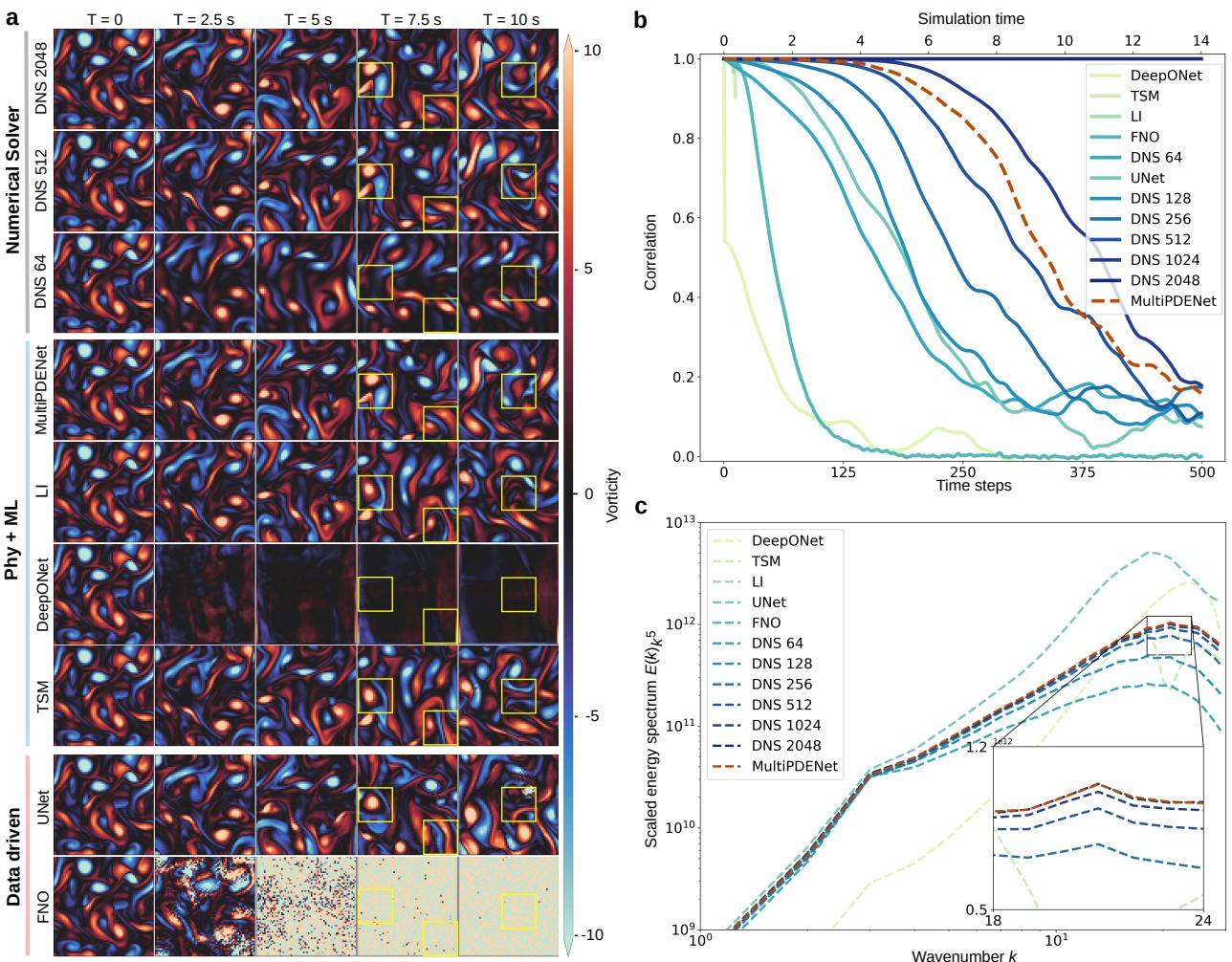

*Figure 4.* Comparison of MultiPDENet and baseline models on Kolmogorov flow with $Re = 1000$. (a) shows the evolution of predicted vorticity fields for reference, MultiPDENet and baselines, starting from the same initial velocities. (b) shows the correlation curve across 500 time steps. (c) shows the scaled energy spectrum scaled by $k^5$ averaged between time steps 100 and 500.

and ground truth trajectories, while Figure 5(**c**) highlights the error distributions, which remain consistently below 0.1, indicating a low error level.

**Test on external forces.** Next, we performed the generalization test using 4 distinct **f**. As shown in Figure 5(**d**), MultiPDENet accurately predicts the trajectories for all forces. Notably, the downward trend in the correlation curve (in Figure 5 (**e**)) for $\mathbf{f}_2$ parallels the behavior observed in Figure 4(**b**), likely because **f** alters only the periodic function without changing the wave number. The error analysis in Figure 5(**f**) confirms that the error levels remain below 0.1.

**Test on flow with** $Re = 4000$**.** Turbulence at high $Re$'s presents significant challenges for prediction due to its nonlinearity and complex vortex structures. To further demonstrate the superior capability of our model, we conducted an additional experiment with a high $Re = 4000$ (see details in Table S2) with the experimental setup in Section 4.1. After

training, the model was tested on 10 trajectories with new ICs. Appendix Figure 6(**a**) illustrates the snapshots predicted by MultiPDENet over 600 timesteps, demonstrating sustained accuracy even at time step 450. The correlation curve in Appendix Figure 6(**b**) highlights the superiority of our model compared to DNS 1024. The error analysis in Appendix Figure 6(**c**) confirms this performance, with errors consistently below 0.01. These results demonstrate the effectiveness of MultiPDENet for higher Reynolds number, e.g., $Re = 4000$, within domain $(0, 2\pi)^2$.

**Test on flow within larger domains.** We extended the spatial domain from $(0, 2\pi)^2$ to $(0, 4\pi)^2$ to further evaluate our model's generalizability over larger mesh grids in a more complex scenario. Larger domains introduce diverse physical phenomena, challenging the model to capture global and local dynamics on coarse grids. Using the same $64 \times 64$ grid, we tested our trained model on 10 unseen trajectories

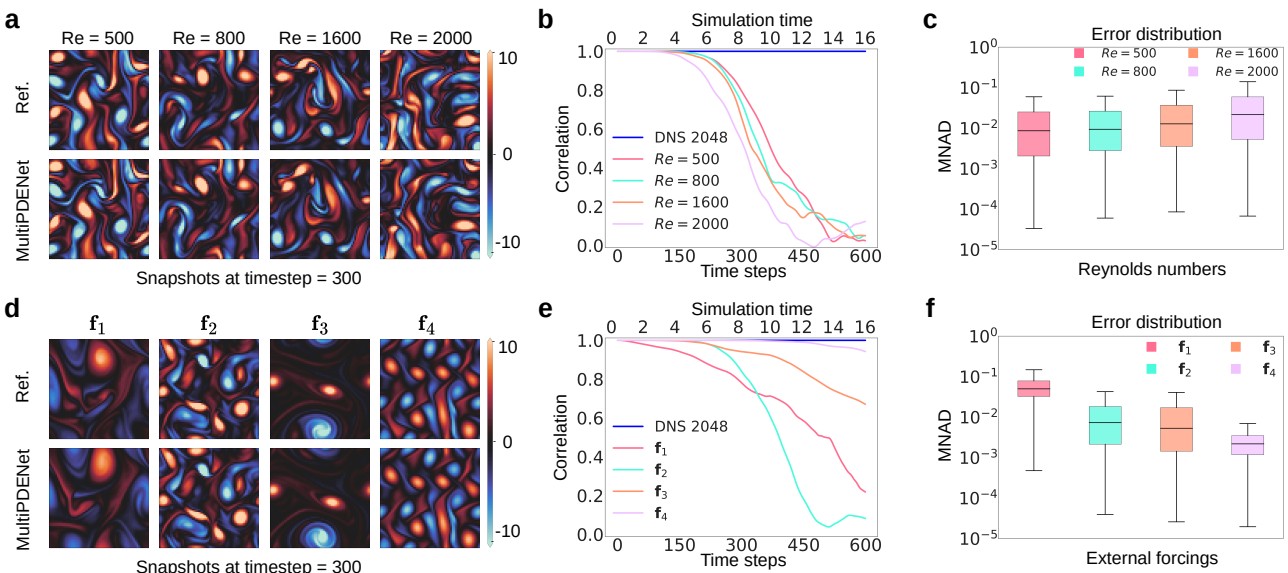

*Figure 5.* MultiPDENet can generalize to simulate different Reynolds numbers and external forcings without retraining. Vorticity snapshots predicted by MultiPDENet and ground truth at timestep = 300 (left), correlation curve over 600 timesteps (middle), error distribution (right). (a-c) display results for different Reynolds numbers, (d-f) show results for varying external forcings.

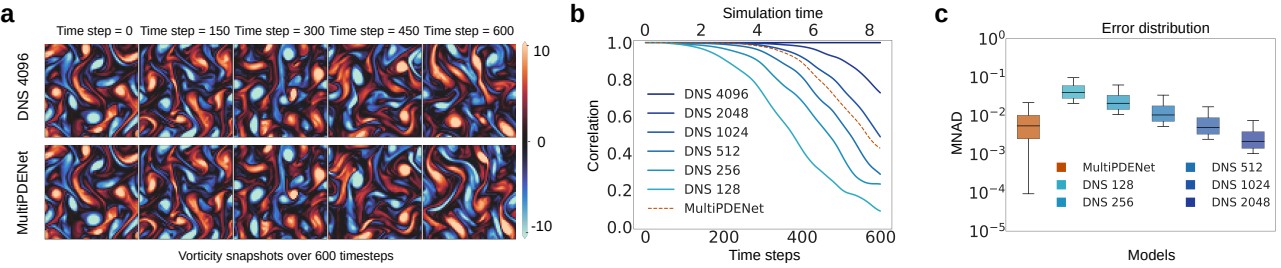

*Figure 6.* MultiPDENet is applicable to high Reynolds number turbulence. (a) Trajectories predicted by MultiPDENet at $Re = 4000$. (b-c) Correlation and error distribution comparison between MultiPDENet and numerical method.

(details in Appendix C). As shown in Appendix Figure 7(**a**), the snapshots over 300 time steps remain accurate. The correlation curve in Appendix Figure 7(**b**) depicts our model's performance closely matches or exceeds DNS 1024. The error distribution in Appendix Figure 7(**c**) shows error levels comparable to DNS 1024. Notably, our model achieves a speedup of 6× compared to the FV DNS method.

In summary, MultiPDENet demonstrates remarkable generalizability, showcasing its ability to capture the underlying dynamics across multiple temporal scales. The embedded learnable PDE module within our model is crucial for enabling robust and accurate predictions.

### 4.4. Ablation Study

To quantify the contribution of each module, we conducted ablation experiments on the KF dataset. Specifically, we compared the following model variations: (1) Model A (no

Poisson Block); (2) Model B (no filter structure constraint); (3) Model C (only Physics Block for prediction); (4) Model D (FD convolution instead of symmetric filter); (5) Model E (no Correction Block); (6) Model F (no $M_iNN$ Block); (7) Model G (no $M_aNN$ Block); (8) Model H (no Physics Block); (9) Model I (forward Euler); and (10) MultiPDENet (the full model). The results are shown in Table 3.

Removing the Poisson Block impairs our model's performance, confirming $p$-**u** decoupling necessity in the NSE. Relaxing the filter structure constraint degrades the result, validating our proposed kernel's efficacy. The Physics Block alone yields worse prediction compared to the full model. Replacing the Conv kernel with FD stencils results in poorer performance, indicating that fixed-value FD kernels are unsuitable for coarse grids. Omitting the Correction Block also degrades the model prediction, highlighting the necessity of field correction. While the model can still accurately predict up to 5.9 s without the $M_iNN$ Block, the error increases by

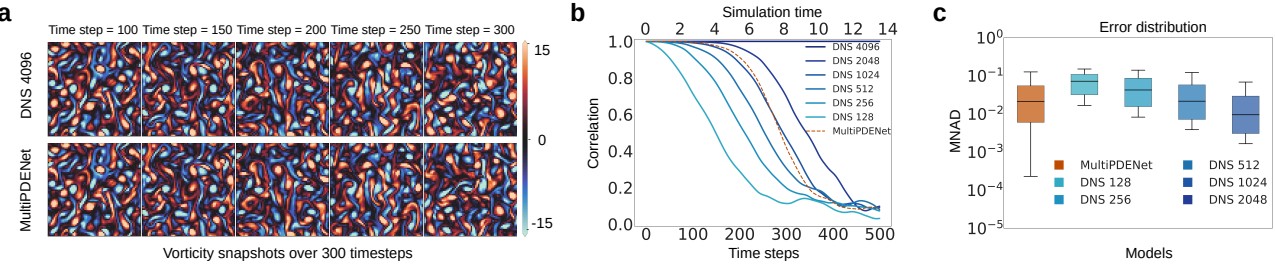

*Figure 7.* MultiPDENet applied to a larger domain. (a) Predicted trajectories within $4\pi \times 4\pi$. (b-c) Correlation and error distribution comparison with the numerical method.

*Table 3.* Results of the ablation study.

| Ablated Model | RMSE ($\downarrow$) | MAE ($\downarrow$) | MNAD ($\downarrow$) | HCT ($\uparrow$) |
|---|---|---|---|---|
| Model A | 0.1601 | 0.0711 | 0.0085 | 7.904 |
| Model B | 0.2432 | 0.1156 | 0.0137 | 7.8633 |
| Model C | 0.2632 | 0.1402 | 0.0186 | 7.3146 |
| Model D | 0.2503 | 0.1410 | 0.0145 | 7.0783 |
| Model E | 0.2958 | 0.1453 | 0.0180 | 6.6856 |
| Model F | 0.4338 | 0.2401 | 0.0285 | 5.9768 |
| Model G | NaN | NaN | NaN | 1.4193 |
| Model H | 1.2023 | 0.9256 | 0.1122 | 0.6227 |
| Model I | 0.4357 | 0.2321 | 0.0278 | 6.2481 |
| MultiPDENet | **0.1379** | **0.0648** | **0.0077** | **8.3566** |

$4\times$, suggesting the need for the $M_iNN$ Block.

Removing the $M_aNN$ Block restricts the micro-scale accuracy and causes macro-scale instability due to error accumulation. This underscores the importance of the $M_aNN$ Block for long-term stability. The omission of the Physics Block (e.g., solely U-Net) cripples the prediction with limited coarse training data,, proving its critical role. Moreover, the use of RK4 outperforms forward Euler in terms of stability and accuracy. Hence, all the components are essential and contribute meaningfully to the MultiPDENet model.

## 5. Conclusion

We introduce an end-to-end physics-encoded network (aka, MultiPDENet) with multi-scale time stepping for accelerated simulation of spatiotemporal dynamics such as turbulent flows. MultiPDENet consists of a multi-scale temporal learning architecture, a learnable Physics Block for solution prediction at the fine time scale, where trainable symmetric filters are designed for improved derivative approximation on coarse spatial grids. Such a method is capable of long-term prediction on coarse grids given very limited training data (see the data size scaling test in Appendix D.4). MultiPDENet outperforms other baselines through extensive tests on fluid dynamics and reaction-diffusion equations. In particular, such a model excels in generalizability over ICs, Reynolds numbers, and external forces in the turbulent flow experiments. MultiPDENet also exhibits strong stabil-

ity in long-term prediction of turbulent flows, effectively capturing both global and local patterns in larger domains.

We also tested the computational efficiency of trained Multi-PDENet for accelerated flow prediction (more details shown in Appendix G.2). For a certain given accuracy (e.g., correlation $\geq 0.8$), MultiPDENet achieves $\geq 5\times$ speedup compared with GPU-accelerated DNS (Appendix Table S8), e.g., JAX-CFD, where all the tests were performed on a single Nvidia A100 80G GPU. However, MultiPDENet still faces some unresolved challenges. Firstly, the model currently only handles regular grids, due to the limitation of convolution operation used in the model. In the future, we aim to address this issue by incorporating graph neural networks to manage irregular grids. Secondly, the model has only been currently tested on 1D and 2D problems. We will extend it to 3D systems in our future work.

## Acknowledgment

The work is supported by the National Science and Technology Major Project (No. 2022ZD0117804), the National Natural Science Foundation of China (No. 62276269 and No. 92270118), the Beijing Natural Science Foundation (No. 1232009), and the Fundamental Research Funds for the Central Universities (No. 202230265). Our source codes are available in the following GitHub repository: `https://github.com/intell-sci-comput/MultiPDENet`.

## Impact statement

This work aims to develop a PDE-embedded network with multiscale time stepping, integrating numerical schemes and machine learning to accelerate flow simulations. This approach has the potential to benefit various fields, including weather forecasting and turbulent flow prediction. This research is solely intended for scientific purposes and does not present any foreseeable ethical risks.

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

# APPENDIX

## A. Related work

**Numerical Methods.** Numerical methods have been extensively applied to solve PDEs. Approaches such as FD (Thomas, 2013), FE (Zienkiewicz et al., 2005), and FV methods (Moukalled et al., 2016) discretize the continuous domain into mesh grids, transforming PDEs into algebraic equations that can be solved with high accuracy. However, these methods often require fine spatiotemporal grids and substantial computational resources to achieve accurate solutions, particularly in high-dimensional spaces. This leads to two main challenges: (1) the need for repeated computations when conditions change (e.g., ICs); (2) the demand for fast simulations in many industrial applications.

**Machine Learning Methods.** Building on the success of machine learning in fields like natural language processing (Vaswani et al., 2017) and computer vision (He et al., 2016), these techniques have also been applied to solving PDEs. With abundant labeled data, it is possible to train end-to-end models to predict solutions. Representative works include ResNet (Lu et al., 2018), CNN-based models (Bhatnagar et al., 2019; Stachenfeld et al., 2022; Gupta & Brandstetter, 2023), Transformer-based models (Cao, 2021; Geneva & Zabaras, 2022; Liu et al., 2024b; Li et al., 2024b), Graph-based models (Brandstetter et al., 2022b; Wu et al., 2022; Zeng et al., 2023), and pretrained diffusion models (Li et al., 2024a). Many notable neural operators (Lu et al., 2021; Li et al., 2021; Rahman et al., 2023; Bonev et al., 2023; Azizzadenesheli et al., 2024), which learn a mapping between functional spaces, enable the approximation of complex relationships in PDEs. While these methods show promise in learning complex dynamics and approximating solutions, they often require substantial amounts of labeled data.

**Physics-inspired Learning Methods.** Recently, physics-inspired learning methods have demonstrated impressive capabilities in solving PDEs, which can be classified into two categories according to the way of adding prior knowledge: physics-informed and physics-encoded. The physics-informed methods take PDEs and I/BCs as a part of the loss functions (e.g., the family of PINN (Raissi et al., 2019; 2020; Wang et al., 2020; Tang et al., 2024; Wu et al., 2024; Chen et al., 2025), PhySR (Ren et al., 2023)). On the other hand, the physics-encoded methods employ a different approach that preserves the structure of PDEs, ensuring that the model adheres to the given equations to capture the underlying dynamics, e.g., EquNN (Wang et al., 2021), TiGNN (Hernández et al., 2023), PeRCNN (Rao et al., 2022; 2023; Ren et al., 2025), PhyMPGN (Zeng et al., 2025), CiGNN (Mi et al., 2025), and HelmFluid (Xing et al., 2024). In addition, other related studies (Long et al., 2018; 2019; Kossaczká et al., 2021; So et al., 2021) have explored the use of CNN as alternative spatial derivative operators for approximating derivatives and capturing the dynamics of interest.

**Hybrid Learning Methods.** Hybrid learning methods combine the strengths of numerical approaches and NNs to improve prediction accuracy. For efficient modeling of spatiotemporal dynamics, these methods can be trained on coarse grids. Representative methods include FV-based neural methods (Kochkov et al., 2021; Sun et al., 2023), FD-based neural methods (Zhuang et al., 2021; Liu et al., 2024a; Wang et al., 2024), and spectral-based neural methods (Dresdner et al., 2023; Arcomano et al., 2022). While these approaches show efficacy in modeling spatiotemporal dynamics, their representation capacities are often limited by the fixed structure of their numerical components. As a result, most of these models still require sufficiently large amounts of training data.

## B. The Details of MultiPDENet

### B.1. Correction Block

The Correction Block leverages a neural network to refine the coarse solution, with the Fourier Neural Operator (FNO) (Li et al., 2021) as the correction mechanism within this block. FNO functions by decomposing the input field into frequency components, processing each frequency individually, and reconstructing the modified spectral information back into the physical domain via the Fourier transform. This layer-wise update process is expressed as:

$$\mathbf{v}^{l+1}(\tilde{\mathbf{x}}) = \sigma\left(\mathbf{W}^l \mathbf{v}^l(\tilde{\mathbf{x}}) + \left(\mathscr{K}(\phi)\mathbf{v}^l\right)(\tilde{\mathbf{x}})\right), \tag{S1}$$

where $\mathbf{v}^l(\tilde{\mathbf{x}})$ denotes the latent feature map at the $l$-th layer, defined on the coarse grid $\tilde{\mathbf{x}}$. The initial feature map is $\mathbf{v}^0(\tilde{\mathbf{x}}) = \mathscr{P}(\bar{\mathbf{u}}_m^k)$, where $\mathscr{P}$ is a local mapping function that projects $\bar{\mathbf{u}}_m^k$ into a higher-dimensional space. The kernel integral transformation is defined as $\mathscr{K}(\phi)(\mathbf{z}) = \mathtt{iFFT}(\mathbf{R}_\phi \cdot \mathtt{FFT}(\mathbf{z}))$, which applies the Fourier transform, spectral

filtering via $\mathbf{R}_\phi$, convolution in the frequency domain, and the inverse Fourier transform to the latent feature map $\mathbf{z}$. Here, $\phi$ represents the trainable parameters, $\sigma(\cdot)$ is the GELU activation function, and $\mathbf{W}^l$ denotes the weights of the linear layer. After passing through an $L$-layer FNO, the refined coarse solution is computed as $\hat{\mathbf{u}}_m^k = \mathcal{Q}(\mathbf{v}^L(\tilde{\mathbf{x}}))$, where $\mathcal{Q}$ projects the latent representation of the final layer back into the original solution space.

In the Correction Block, we set $L = 2$. For the Burgers equation, we configure the model with modes = 12, width = 12, and a projection from 12 channels to 50 channels. For the GS case, we use the same configuration: modes = 12, width = 20, with a projection from 20 channels to 50 channels. For the KdV equation, the setup is defined as modes = 32 and width = 64, with a projection from 64 channels to 128 channels. The NSE case, however, requires a different setup: modes = 25, width = 20, with a projection from channel = 20 to channel = 128.

### B.2. Physics Block

To accurately predict at the micro-scale step, we developed a neural solver called the Physics Block, ensuring stability, accuracy, and efficiency through adherence to the Courant-Friedrichs-Lewy (CFL) conditions (LeVeque, 2007). The Physics Block comprises three key components: the Poisson Block (Figure S1(**a**)-**b**)), the PDE Block (Figure 1(**c**)), and the M$_i$NN Block (Figure S1(**c**)).

**Poisson Solver.** The pressure field is computed using the spectral method, which involves solving the Poisson equation:

$$\Delta p = \psi(\mathbf{u}). \tag{S2}$$

Here, $\psi(\mathbf{u}) = 2\left(u_x v_y - u_y v_x\right)$ represents the source term for the pressure.

Applying the Fast Fourier Transform (`FFT`) to Eq. (S2), we obtain:

$$-(\varphi_x^2 + \varphi_y^2)p^* = \psi^*(\mathbf{u}), \tag{S3}$$

where $\varphi_x$ and $\varphi_y$ are the wavenumbers in the $x$ and $y$ directions, respectively. Assuming $\varphi_x^2 + \varphi_y^2 \neq 0$, we can solve for the pressure in the frequency domain:

$$p^* = \frac{\psi^*(\mathbf{u})}{-(\varphi_x^2 + \varphi_y^2)}. \tag{S4}$$

Finally, the pressure field is recovered in the spatial domain using the inverse FFT (`iFFT`):

$$p = \texttt{iFFT}\left[p^*\right]. \tag{S5}$$

This spectral method offers an efficient approach to calculating the pressure field without the need for labeled data or training.

**BC encoding.** To ensure that the solution obeys the given periodic boundary conditions and that the feature map shape remains unchanged after differentiation, we employ periodic BC padding (see Figure S2) in our architecture. This method of hard encoding padding not only guarantees that the boundary conditions are periodic, but also improves accuracy.

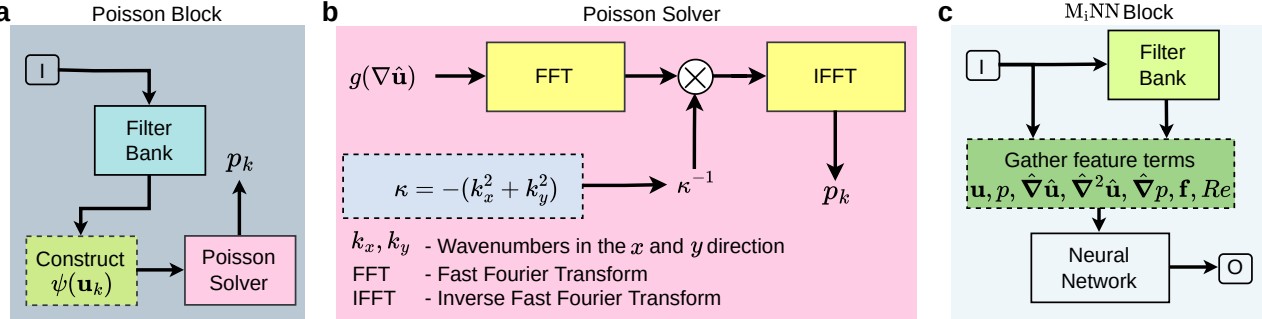

*Figure S1.* Components of Physics Block. (a), Poisson block. (b), Poisson solver. (c), M$_i$NN block.

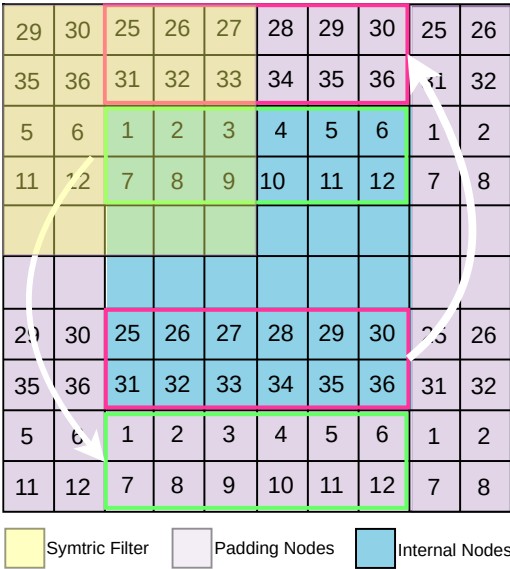

*Figure S2.* Periodic BC padding.

## B.3. RK4 Integration Scheme

RK4 is a widely used numerical integration method for solving ordinary differential equations (ODEs) and PDEs, commonly employed as a time integration solver. It provides a balance between computational efficiency and accuracy by calculating intermediate slopes at various points within each time step. The general numerical integration method for time marching from $\mathbf{u}_{t_j}$ to $\mathbf{u}_{t_{j+1}}$ can be written as:

$$\mathbf{u}_{j+1} = \mathbf{u}_j + \int_{t_j}^{t_{j+1}} \mathcal{B}(\mathbf{u}_j(\tilde{\mathbf{x}}, \tau))\mathrm{d}\tau. \tag{S6}$$

Among them, $\mathbf{u}_{j+1}$ and $\mathbf{u}_j$ are solutions at time $j + 1$ and $j$. RK4 is a high-order integration scheme, which divides the time interval into multiple equally spaced small time steps to approximate the integral. The final update of the above state change can be written as:

$$
\begin{aligned}
\mathbf{r}_1 &= \mathcal{B}\left(\mathbf{u}_j, t_j\right), \\
\mathbf{r}_2 &= \mathcal{B}\left(\mathbf{u}_j + \frac{\delta t}{2} \times \mathbf{r}_1, t_j + \frac{\delta t}{2}\right), \\
\mathbf{r}_3 &= \mathcal{B}\left(\mathbf{u}_j + \frac{\delta t}{2} \times \mathbf{r}_2, t_j + \frac{\delta t}{2}\right), \\
\mathbf{r}_4 &= \mathcal{B}\left(\mathbf{u}_j + \delta t \times \mathbf{r}_3, t_j + \delta t\right), \\
\mathbf{u}_{j+1} &= \mathbf{u}_j + \frac{1}{6}\delta t(\mathbf{r}_1 + 2\mathbf{r}_2 + 2\mathbf{r}_3 + \mathbf{r}_4),
\end{aligned}
\tag{S7}
$$

where $\delta t$ denotes the step size and $\mathbf{r}_1, \mathbf{r}_2, \mathbf{r}_3, \mathbf{r}_4$ represent four intermediate variables (slopes). The global error is proportional to the step size to the fourth power, i.e., $\mathcal{O}(\delta t^4)$.

## C. Data Details

**KdV.** The Korteweg-de Vries (KdV) equation is a well-known nonlinear PDE used to describe the movement of shallow water waves with small amplitude in a channel. It models the dynamics of these waves and is particularly noted for its ability to represent solitary waves, or solutons, given by:

$$\frac{\partial u}{\partial t} + u\frac{\partial u}{\partial x} + \frac{\partial^3 u}{\partial x^3} = 0, \tag{S8}$$

*Table S1.* Settings for generating datasets.

| Parameters / Case | KdV | Burgers | GS | NSE |
|---|---|---|---|---|
| DNS Method | Spectral | FD | FD | FV |
| Spatial Domain | $[0, 64]$ | $[0, 1]^2$ | $[0, 1]^2$ | $[0, 2\pi]^2$ |
| Calculate Grid | 256 | $100^2$ | $128^2$ | $2048^2$ |
| Training Grid | 64 | $25^2$ | $32^2$ | $64^2$ |
| Simulation $dt$ (s) | $1.00 \times 10^{-2}$ | $1.00 \times 10^{-3}$ | $2.00 \times 10^{-3}$ | $2.19 \times 10^{-4}$ |
| Warmup (s) | 0 | 0.1 | 0 | 40 |
| Training data group | 3 | 5 | 3 | 5 |
| Testing data group | 10 | 10 | 10 | 10 |
| Spatial downsample | $4\times$ | $16\times$ | $16\times$ | $1024\times$ |
| Temporal downsample | $5\times$ | $10\times$ | $20\times$ | $128\times$ |

where $u = u(x, t)$ represents the wave amplitude as a function of position $x$ and time $t$. $u\frac{\partial u}{\partial x}$ accounts for the nonlinear effects, while $\frac{\partial^3 u}{\partial x^3}$ captures the dispersive effects in the system. The equation illustrates how a balance between these two effects leads to the formation of solitary waves, or solitons, that maintain their shape over long distances.

To generate the dataset, we employ the method of lines (MOL) using pseudospectral methods to compute the spatial derivatives (Brandstetter et al., 2022a). The dataset is initially generated on a grid of 256 points and then downsampled to a grid of 64 points for numerical experiments. The simulation timestep is set to $dt = 1 \times 10^{-2}$ seconds, with the total simulation duration set to 100 seconds. For training, we use 3 sets of data, each comprising 1000 timesteps over $\Delta t = 5dt$, along with ten additional testing sets with different ICs.

**Burgers.** This equation models the behavior of a viscous fluid (Kumar, 2023), incorporating both nonlinear dynamics and diffusion effects. It finds extensive applications across various scientific disciplines, including fluid mechanics, materials science, applied mathematics and engineering. The equation is expressed as follows:

$$\frac{\partial \mathbf{u}}{\partial t} = \nu \boldsymbol{\nabla}^2 \mathbf{u} - \mathbf{u} \cdot \boldsymbol{\nabla} \mathbf{u}, \quad t \in [0, T], \tag{S9}$$

where $\mathbf{u} = \{u, v\} \in \mathbb{R}^2$ represents the fluid velocities, $\nu$ is the viscosity coefficient set to 0.002, and $\Delta$ is the Laplacian operator.

As shown in Table S1, we generate the dataset using the finite difference method with a 4th-order Runge–Kutta time integration (Rao et al., 2023) and periodic boundary conditions over the spatial domain $\mathbf{x} \in [0, 1]$. The data is initially generated on a $100^2$ grid and subsequently downsampled to a $25^2$ grid for use in numerical experiments. The simulation timestep is set to $dt = 1 \times 10^{-3}$ seconds, with a total duration of $T = 1.4$ seconds. During the training stage, we employ five trajectories with $\Delta t = 10\delta t$, each consisting of 140 snapshots. In the testing stage, we use ten different trajectories, each containing 140 snapshots.

**GS.** The Gray-Scott (GS) reaction-diffusion model is a system of PDEs that describes the interaction and diffusion of two reacting chemicals. It is known for its ability to produce intricate and evolving patterns, making it a popular model for studying pattern formation. It is widely used in fields such as chemistry, biology, and physics to simulate processes like chemical reactions and biological morphogenesis. The equation is expressed by:

$$\begin{aligned}
\frac{\partial u}{\partial t} &= D_u \Delta u - uv^2 + \alpha(1 - u), \\
\frac{\partial v}{\partial t} &= D_v \Delta v + uv^2 - (\alpha + \kappa)v,
\end{aligned} \tag{S10}$$

where $u$ and $v$ denote the concentrations of two distinct chemical species, with $D_u$ and $D_v$ indicating their respective diffusion coefficients. The first equation models the change in the concentration of $u$ over time. The term $D_u \Delta u$ represents the diffusion of $u$, $-uv^2$ describes the reaction between $u$ and $v$, and $\alpha(1 - u)$ represents the replenishment of $u$ based on the feed rate $\alpha$. The second equation models the evolution of $v$, where $D_v \Delta v$ accounts for diffusion, $uv^2$ represents the creation of $v$ from the reaction with $u$, and $-(\alpha + \kappa)v$ describes the decay of $v$, with $\kappa$ as the decay rates.

*Table S2.* Settings for generating the NSE datasets.

| Dataset | Grid | Spatial Domain | $Re$ | Warmup time | dt | Innerstep |
|---------|------|----------------|------|-------------|-----|-----------|
| $\mathbf{f_1} \sim \mathbf{f_4}$ | $2048^2 \rightarrow 64^2$ | $(0, 2\pi)^2$ | 1000 | 40 | $2.1914 \times 10^{-4}$ | 32 |
| $Re = 500$ | $2048^2 \rightarrow 64^2$ | $(0, 2\pi)^2$ | 500 | 40 | $2.1914 \times 10^{-4}$ | 32 |
| $Re = 800$ | $2048^2 \rightarrow 64^2$ | $(0, 2\pi)^2$ | 800 | 40 | $2.1914 \times 10^{-4}$ | 32 |
| $Re = 1000$ | $2048^2 \rightarrow 64^2$ | $(0, 2\pi)^2$ | 1000 | 40 | $2.1914 \times 10^{-4}$ | 32 |
| $Re = 1600$ | $2048^2 \rightarrow 64^2$ | $(0, 2\pi)^2$ | 1600 | 40 | $2.1914 \times 10^{-4}$ | 32 |
| $Re = 2000$ | $2048^2 \rightarrow 64^2$ | $(0, 2\pi)^2$ | 2000 | 40 | $2.1914 \times 10^{-4}$ | 32 |
| $Re = 4000$ | $4096^2 \rightarrow 64^2$ | $(0, 2\pi)^2$ | 4000 | 40 | $1.0957 \times 10^{-4}$ | 32 |
| $Re = 1000$ | $4096^2 \rightarrow 64^2$ | $(0, 4\pi)^2$ | 1000 | 40 | $1.0957 \times 10^{-4}$ | 32 |

We also utilize the RK4 time integration method for dataset generation. In this case, we assign the values $D_u = 2.0 \times 10^{-5}$, $D_v = 5.0 \times 10^{-6}$, $\alpha = 0.04$, and $\kappa = 0.06$. The dataset is generated using the finite difference method on a $128^2$ grid with periodic boundary conditions, spanning the spatial domain $\mathbf{x} \in [0, 1]^2$. To generate different ICs, we first define a grid based on the spatiotemporal resolution and initialize the concentrations of two chemicals. By setting different random seeds and adding varied random noise, we create unique ICs. The simulation uses a timestep of $dt = 0.5$ s over a total duration of $T = 1400$ seconds. The data is then downsampled to a $32^2$ grid, and the timestep is increased to 10 seconds ($\Delta t = 20dt$) for ground truth creation. We utilize three training trajectories, each with 180 snapshots, and ten additional testing sets with varying ICs.

**NSE.** The Navier-Stokes equations (NSE) are fundamental to the study of fluid dynamics, governing the behavior of fluid motion. In this paper, we focus on a two-dimensional, incompressible Kolmogorov flow with periodic boundary conditions, expressed in velocity-pressure form as:

$$\frac{\partial \mathbf{u}}{\partial t} + (\mathbf{u} \cdot \boldsymbol{\nabla})\mathbf{u} = \frac{1}{Re}\boldsymbol{\nabla}^2\mathbf{u} - \boldsymbol{\nabla}p + \mathbf{f}, \quad t \in [0, T],$$
$$\boldsymbol{\nabla} \cdot \mathbf{u} = 0, \tag{S11}$$

where $\mathbf{u} = \{u, v\} \in \mathbb{R}^2$ denotes the fluid velocity vector, $p \in \mathbb{R}$ represents the pressure, and $Re$ is the Reynolds number that characterizes the flow regime. The Reynolds number serves as a scaling factor in the NSE, balancing the inertial forces, represented by the advection term $(\mathbf{u} \cdot \boldsymbol{\nabla})\mathbf{u}$, with the viscous forces, captured by the Laplacian term $\Delta\mathbf{u}$. When $Re$ is low, the flow remains predominantly laminar and smooth due to the dominance of the viscous forces. Conversely, at high Reynolds numbers, the inertial forces take precedence, leading to a more chaotic and turbulent flow behavior.

To create the dataset, we follow the approach outlined in JAX-CFD (Kochkov et al., 2021). We simulate data using the Finite Volume Method (FVM) on a fine grid with a time step of $dt$ (e.g., $Re = 1000$, $2048 \times 2048$). This data is then downsampled to a coarse grid with $\Delta t = 128dt$ (e.g., $Re = 1000$, $64 \times 64$) to serve as the ground truth. Different ICs are generated by introducing random noise into each component of the velocity field and subsequently filtering it to obtain a divergence-free field with the desired properties. For training, we utilize only five groups of labeled data with 4800 snapshots, while testing involves ten sets of trajectories. The model performance tests include trajectories with different Reynolds numbers $Re = 500, 800, 1600, 2000, 4000$, different external forces $\mathbf{f}_1 = \cos(2y)\boldsymbol{\eta}_x - 0.1\mathbf{u}$, $\mathbf{f}_2 = 0$ $\mathbf{f}_3 = \cos(4y)\boldsymbol{\eta}_x - 0.1\mathbf{u}$, $\mathbf{f}_4 = \sin(4y)\boldsymbol{\eta}_x - 0.4\mathbf{u}$, and a larger computational domain $\mathbf{x} \in (0, 4\pi)^2$. The detailed dataset parameters are shown in Table S2.

## D. Additional Experimental Results

### D.1. Less Data and Added Noise

To evaluate our model's robustness against missing data and noise, we conducted experiments on the NSE using five sets of trajectories ($5 \times 1200 \times 2 \times 64 \times 64$). We tested two conditions: (1) randomly removing 20% of snapshots and (2) adding 0.1% Gaussian noise during training. As shown in Table S3, the model's performance was only slightly impacted in Experiment 1, with low error rates. In Experiment 2, HCT remained above 8 s. These results highlight the model's strong generalization ability even under challenging conditions.

*Table S3.* Performance metrics under different noise levels during training.

| Training | RMSE | MAE | MNAD | HCT (s) |
|---|---|---|---|---|
| - 20% data | 0.1935 | 0.0958 | 0.0113 | 8.1392 |
| + 0.1% noise | 0.2083 | 0.1014 | 0.0123 | 8.0431 |
| normal | 0.1379 | 0.0648 | 0.0077 | 8.3566 |

*Table S4.* Performance metrics for different NN blocks.

| Model | RMSE | MAE | MNAD | HCT |
|---|---|---|---|---|
| Model-a | NaN | NaN | NaN | 0.8846 |
| Model-b | NaN | NaN | NaN | 5.2930 |
| Model-c | 0.2575 | 0.1507 | 0.0191 | 7.2930 |
| Model-d | 0.1564 | 0.0703 | 0.0083 | 8.0525 |
| Model-e | 0.2479 | 0.1242 | 0.0197 | 7.6346 |
| **MultiPDENet** | **0.1379** | **0.0648** | **0.0077** | **8.3566** |

### D.2. Parametric Experiments on $M_i$NN Block and $M_a$NN Block

To investigate the role of the NN blocks in our model, we conducted additional comparative experiments with the following configurations:

- Model-a: the $M_i$NN Block was set to UNet and the $M_a$NN Block to FNO;

- Model-b: both the $M_i$NN Block and the $M_a$NN Block were set to FNO;

- Model-c: both blocks were set to FNO with roll-out training applied at the macro step (with an unrolled step size of 8);

- Model-d: the $M_i$NN Block was set to DenseCNN and the $M_a$NN Block to UNet;

- Model-e: the $M_i$NN Block was set to FNO while the $M_a$NN Block was set to Swin Transformer (Liu et al., 2021).

All other experimental settings were kept consistent, and the results are presented in Table S4. Model-a and Model-b encountered NaN values, which can be attributed to the $M_a$NN Block requiring a model capable of robust predictions at the macro step. Without such a model, multi-step roll-out training (as in Model-c) becomes necessary to enhance the model's stability in long-term predictions. When a strong predictive module is employed at the macro step (e.g., UNet), the $M_i$NN Block can be replaced with a more parameter-efficient model, such as DenseCNN (Model-d). Setting the $M_a$NN Block to Swin Transformer resulted in a slight decrease in accuracy, which can be attributed to the relatively small size of our dataset, as the Swin Transformer typically excels on larger datasets.

### D.3. Generalization Test on flow with $Re = 4000$.

To further demonstrate the superior capability of our model, we conducted an additional experiment with a high Reynolds number $Re = 4000$ (see details in Table S2) maintaining the experimental setup of Section 4.1. The result is shown in Figure 6.

### D.4. Scaling against Data Size

As shown in Figure S3, our testing results demonstrate that the model exhibits a scaling law behavior, with the RMSE gradually decreasing as the amount of training data increases. Our model adheres to this scaling law as well. Moreover, even in scenarios with limited data regimes (e.g., only five trajectories), the model achieves a low level of error, highlighting its robustness and ability to learn from limited data.

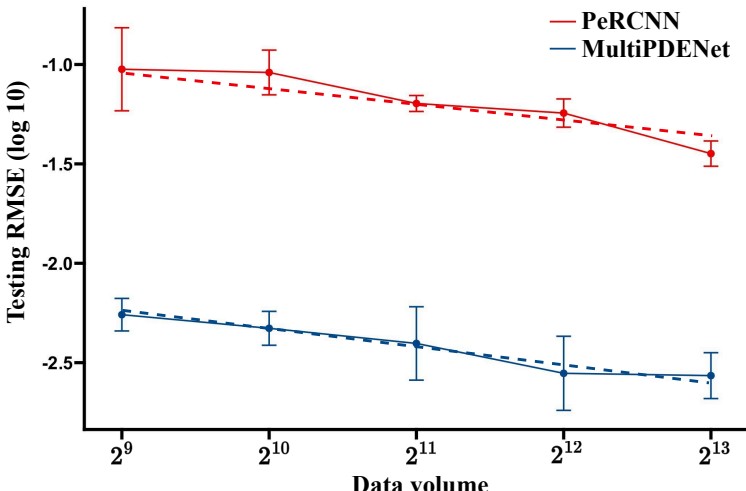

*Figure S3*. Comparison of PeRCNN and MultiPDENet across various training set sizes on the Burgers equation. The $x$-axis represents data volume, defined as the product of trajectory timesteps and the number of trajectories (corresponding to 5, 8, 16, 32, and 64 trajectories, respectively). Dotted lines denotes the linear interpolation.

## E. Evaluation Metrics

We employ several metrics to assess the performance of the tested models, including Root Mean Square Error (RMSE), Mean Absolute Error (MAE), Mean Normalized Absolute Difference (MNAD), and High Correction Time (HCT) (Sun et al., 2023). RMSE quantifies the average magnitude of the errors between predicted and actual values, providing insight into the model's accuracy. MAE assesses the average absolute deviation between predicted and observed values, thereby indicating the scale of the errors. MNAD serves as an important metric for evaluating the consistency of model outputs over time, calculating the average discrepancy across temporal data points and offering a normalized measure of prediction error relative to the range of actual data. HCT gauges the model's capability for making reliable long-term predictions. These metrics are defined as follows:

$$\text{RMSE} = \sqrt{\frac{1}{n} \sum_{i=1}^{n} \|\mathbf{H}_i - \check{\mathbf{H}}_i\|^2}, \tag{S12}$$

$$\text{MAE} = \frac{1}{n} \sum_{i=1}^{n} \left| \mathbf{H}_i - \check{\mathbf{H}}_i \right|, \tag{S13}$$

$$\text{MNAD} = \frac{1}{n} \sum_{i=1}^{n} \frac{\|\mathbf{H}_i - \check{\mathbf{H}}_i\|}{\|\mathbf{H}_i\|_{\max} - \|\mathbf{H}_i\|_{\min}}, \tag{S14}$$

$$\text{HCT} = \sum_{i=1}^{N} \Delta t \cdot [PCC(\mathbf{H}_i, \check{\mathbf{H}}_i) > 0.8], \tag{S15}$$

where

$$PCC(\mathbf{H}_i, \check{\mathbf{H}}_i) = \frac{\text{cov}(\mathbf{H}_i, \check{\mathbf{H}}_i)}{\sigma_{\mathbf{H}_i} \sigma_{\check{\mathbf{H}}_i}}. \tag{S16}$$

Here, $n$ represents the number of trajectories; $\mathbf{H}_i$ denotes the ground truth for each trajectory; $\check{\mathbf{H}}_i$ indicates the spatiotemporal sequence predicted by the model. The term "cov" refers to the covariance function, while "$\sigma$" represents the standard deviation of the respective sequence. The Iverson bracket returns a value of 1 when the condition ($PCC(\mathbf{H}_i, \check{\mathbf{H}}_i) > 0.8$) is satisfied and 0 otherwise. The variable $N$ signifies the total number of time steps.

# F. Baseline Models

**Fourier Neural Operator (FNO).** The FNO (Li et al., 2021) combines neural networks with Fourier transforms to effectively capture both global and local features of system dynamics. The architecture has two key components: First, it applies Fourier transforms to the system state, performing convolution operations in the frequency domain to capture global information. This is followed by an inverse Fourier transform which maps the data back to the spatial domain. Secondly, convolutional layers are employed to extract local features directly from the system state. The outputs from both global and local components are then integrated through the application of activation functions, which ultimately yield the final prediction.

**PINO.** PINO (Li et al., 2024c) shares the same architecture as FNO but incorporates physics-based constraints by embedding governing equations into the loss function. The loss function can be expressed as: $\mathcal{L}(\lambda) = \mathcal{L}_{Eq} + \mathcal{L}_{Data}$. The first term of the loss function, $\mathcal{L}_{Eq}$, represent an MSE loss computed using the analytical expressions of the dynamics, as defined in Equation 1. The second term $\mathcal{L}_{Data}$ is the data loss, which was previously discussed in Section 4.1.

**PeRCNN.** PeRCNN (Rao et al., 2023) integrates physics-based principles directly into the learning framework by embedding governing equations into the neural network structure. The architecture features multiple parallel CNNs that model polynomial relationships via feature map multiplications. This incorporation of physical laws improves the model's generalization and extrapolation capabilities, enabling accurate predictions in dynamic systems governed by complex equations.

**UNet.** The UNet architecture (Ronneberger et al., 2015) adopts a symmetric encoder-decoder structure originally designed for computer vision tasks. The encoder compresses the input by applying multiple downsampling layers to capture hierarchical features at various scales. Conversely, the decoder gradually restores the original spatial resolution using upsampling operations. Skip connections bridge the encoder and decoder, directly transferring feature maps to retain fine-grained details. This design allows UNet to merge high-resolution spatial information with deeper, abstract features, achieving accurate reconstructions of both local and global structures.

**DeepONet.** DeepONet (Lu et al., 2021) is designed to approximate operators and map inputs directly to outputs by leveraging neural networks. The architecture consists of two main components: the trunk network, which processes domain-specific information, and the branch network, which handles the input functions. This dual-structure approach enables the efficient learning of complex functional relationships and enhances the model's capability to capture detailed operator mappings across various applications.

**Learned Interpolation (LI).** The LI (Kochkov et al., 2021) employs a finite volume approach enhanced with neural networks as a replacement for conventional polynomial-based interpolation schemes in computing velocity tensor product. The network adapts to the local flow conditions by learning a dynamic interpolation mechanism that can adjust to the characteristics of the flow. This enables LI to provide accurate fluid dynamics predictions even on coarse grids, improving computational efficiency while maintaining prediction fidelity.

**Temporal Stencil Modeling (TSM).** TSM (Sun et al., 2023) addresses time-dependent partial differential equations (PDEs) in conservation form by integrating time-series modeling with learnable stencil techniques. It effectively recovers information lost during downsampling, enabling enhanced predictive accuracy. TSM is particularly advantageous for machine learning models dealing with coarse-resolution datasets.

# G. Computational Details

## G.1. Training Details

All experiments (both training and inference) in this study were conducted on a single Nvidia A100 GPU (with 80GB memory) running on a server with an Intel(R) Xeon(R) Platinum 8380 CPU (2.30GHz, 64 cores). All model training efforts were performed on coarse grids (see Table 1).

**MultiPDENet.** The MultiPDENet architecture employs the Adam optimizer with a learning rate of $5 \times 10^{-3}$. The model is trained over 1000 epochs with a batch size of 90. Detailed settings for the rollout timestep can be found in Table 1. Additionally, we use the StepLR scheduler to adjust the learning rate by a factor of 0.96 every 200 steps. The model hyperparameters are listed in Tables S5 and S6.

*Table S5.* Overview of hyperparameters used in the M$_i$NN Block.

| Case | Hyperparameters | Value |
|---|---|---|
| NSE | Network | FNO (Li et al., 2021) |
| | Layers | 6 |
| | Modes | 30 |
| | Width | 30 |
| | Blocks | 1 |
| | Padding | periodical |
| | $\sigma$ | GELU |
| | Inputs | $\{\bar{\mathbf{u}}_m^k, \Xi_m^k(p, \hat{\boldsymbol{\nabla}}\hat{\mathbf{u}}, \hat{\boldsymbol{\nabla}}^2\hat{\mathbf{u}}, \hat{\boldsymbol{\nabla}}p, \mathbf{f}, Re)\}$ |
| Burgers | Network | FNO (Li et al., 2021) |
| | Layers | 4 |
| | Modes | 12 |
| | Width | 12 |
| | Blocks | 1 |
| | Padding | periodical |
| | $\sigma$ | GELU |
| | Inputs | $\{\bar{\mathbf{u}}_m^k\}$ |
| GS | Network | FNO (Li et al., 2021) |
| | Layers | 6 |
| | Modes | 12 |
| | Width | 22 |
| | Blocks | 1 |
| | Padding | periodical |
| | $\sigma$ | GELU |
| | Inputs | $\{\bar{\mathbf{u}}_m^k\}$ |
| KdV | Network | FNO (Li et al., 2021) |
| | Layers | 4 |
| | Modes | 32 |
| | Width | 64 |
| | Blocks | 1 |
| | Padding | periodical |
| | $\sigma$ | ReLU |
| | Inputs | $\{\bar{\mathbf{u}}_m^k\}$ |

**FNO.** The architecture of the FNO network closely follows that presented in the original study (Li et al., 2021), with the main adjustment being the adaptation of its training methodology to an autoregressive framework. The training utilizes the Adam optimizer with a learning rate of $1 \times 10^{-3}$ and a batch size of 20. Training is carried out for 1000 epochs, and the rollout timestep matches MultiPDENet.

**UNet.** We implement the modern UNet architecture (Gupta & Brandstetter, 2023) using its default settings, ensuring that the rollout timestep is consistent with that of the MultiPDENet. The StepLR scheduler is employed with a step size of 100 and a gamma of 0.96. The optimizer is Adam, with a learning rate of $1 \times 10^{-3}$ and a batch size of 10. The model is trained for 1000 epochs.

**DeepONet.** We utilize the default configuration of DeepONet (Lu et al., 2021) along with the Adam optimizer. The learning rate is established at $5 \times 10^{-4}$, with a decay factor of 0.9 applied every 5000 steps. The model is trained using a batch size of 16 over a total of 20000 epochs.

**PeRCNN.** We maintain the standard architecture of PeRCNN (Rao et al., 2023). The optimization process is executed with the Adam optimizer and employs a StepLR scheduler that reduces the learning rate by a factor of 0.96 every 100 steps. The initial learning rate is set to 0.01, and the training is conducted over 1000 epochs with a batch size of 32.

*Table S6.* Overview of hyperparameters used in the M$_a$NN Block.

| Case | Hyperparameters | Value |
|------|-----------------|-------|
| NSE | Network | U-Net (Gupta & Brandstetter, 2023) |
| | Hidden size | [128, 128, 256, 512] |
| | Blocks | 2 |
| | Padding | periodical |
| | Inputs | $\{\mathbf{u}, \Delta t, dx\}$ |
| | $\sigma$ | GELU |
| Burgers | Network | U-Net (Gupta & Brandstetter, 2023) |
| | Hidden size | [64, 64, 128, 256] |
| | Blocks | 2 |
| | Padding | periodical |
| | Inputs | $\{\mathbf{u}, \Delta t, dx\}$ |
| | $\sigma$ | GELU |
| GS | Network | U-Net (Gupta & Brandstetter, 2023) |
| | Hidden size | [64, 64, 128, 256] |
| | Blocks | 2 |
| | Padding | periodical |
| | Inputs | $\{\mathbf{u}, \Delta t, dx\}$ |
| | $\sigma$ | GELU |
| KdV | Network | U-Net (Gupta & Brandstetter, 2023) |
| | Hidden size | [64, 64, 128, 256] |
| | Blocks | 2 |
| | Padding | periodical |
| | Inputs | $\{\mathbf{u}, \Delta t, dx\}$ |
| | $\sigma$ | ReLU |

**LI.** We adopt the default network architecture and parameter settings for LI (Kochkov et al., 2021). The optimizer used is Adam with $\beta_1 = 0.9$ and $\beta_2 = 0.99$. The batch size is configured to 8, along with a global gradient norm clipping threshold of 0.01. The learning rate is set to $1 \times 10^{-3}$, and weight decay is configured to $1 \times 10^{-6}$.

**TSM.** We follow the default network architecture and parameter settings for TSM (Sun et al., 2023). The initial learning rate is set to $1 \times 10^{-4}$, with a weight decay of $1 \times 10^{-4}$. The gradient clipping norm is configured to be $1 \times 10^{-2}$. We use the Adam optimizer with $\beta_2 = 0.98$, and the batch size is set to 8.

### G.2. Computational Cost (Inference)

Taking NSE as an example, we compared the inference time, RMSE, and HCT of MultiPDENet with the Direct Numerical Simulation (DNS) method across three cases. The comparison principle is based on the time required to simulate the same trajectory length ($T = 8.4$ s) under identical experimental conditions (a single A100 GPU). The inference time is measured from the moment the initial conditions (IC) are fed into the model until the trajectory of the same length is predicted.

The DNS settings follow JAX-CFD (Kochkov et al., 2021). According to the CFL condition, the simulated time step ($dt$) varies with the resolution of the DNS method, resulting in different numbers of timesteps required for calculation. DNS 2048, DNS 4096, and DNS 4096 are used as the ground truth for the three cases, respectively. Detailed comparison results are presented in Table S7. The computational time for a given accuracy (e.g., correlation $\geq 0.8$) on the NSE dataset is depicted in Table S8.

Nevertheless, we also would like to clarify that the DNS code used above was implemented in JAX, while our model was programmed in PyTorch. These two platforms have distinct efficiencies even for the same model. Typically, the codes under JAX environment runs much faster compared with PyTorch (up to $6\times$) (Takamoto et al., 2022). We anticipate to achieve much higher speedup of our model if also implemented and optimized in JAX, which is, however, out of the scope of the present study.

*Table S7.* Performance comparison of different methods on the NSE dataset across various cases.

| Case | Method | Timestep | Infer Cost (s) | RMSE | HCT (s) |
|---|---|---|---|---|---|
| $Re = 1000$ | DNS 2048 | 38400 | 260 | 0 | 8.4 |
| $Re = 1000$ | DNS 1024 | 19200 | 135 | 0.1267 | 8.4 |
| $Re = 1000$ | DNS 512 | 9600 | 52 | 0.2674 | 6.5 |
| $Re = 1000$ | DNS 64 | 1200 | 18 | 0.7818 | 2.7 |
| $Re = 1000$ | **MultiPDENet** | 300 | 26 | 0.1379 | 8.4 |
| $Re = 4000$ | DNS 4096 | 76800 | 1400 | 0 | 8.4 |
| $Re = 4000$ | DNS 1024 | 19200 | 136 | 0.1463 | 6.8 |
| $Re = 4000$ | DNS 512 | 9600 | 52 | 0.2860 | 5.8 |
| $Re = 4000$ | DNS 128 | 2400 | 31 | 0.8658 | 3.6 |
| $Re = 4000$ | **MultiPDENet** | 300 | 26 | 0.1685 | 6.4 |
| $x \in (0, 4\pi)^2$ | DNS 4096 | 75750 | 1280 | 0 | 8.4 |
| $x \in (0, 4\pi)^2$ | DNS 1024 | 19200 | 129 | 0.4638 | 6.6 |
| $x \in (0, 4\pi)^2$ | DNS 512 | 9600 | 50 | 0.6166 | 5.2 |
| $x \in (0, 4\pi)^2$ | DNS 128 | 2400 | 30 | 0.8835 | 2.3 |
| $x \in (0, 4\pi)^2$ | **MultiPDENet** | 300 | 26 | 0.4577 | 6.7 |

*Table S8.* Computational time for a given accuracy (e.g., correlation $\geq 0.8$) on the NSE dataset.

| Iterm | $Re = 1000$ | $Re = 4000$ | $\mathbf{x} \in [0,4\pi]^2$ |
|---|---|---|---|
| DNS 1024 | 135 s | 130 s | 133 s |
| MultiPDENet | 26 s | 19 s | 21 s |
| Speed up | **5×** | **7×** | **6×** |

