# OpenReview forum: "MultiPDENet: PDE-embedded Learning with Multi-time-stepping for Accelerated Flow Simulation"
_ICML.cc/2025/Conference — ICML 2025 poster_

### Official Review · Reviewer_XEPN · 2025-03-07

**Overall Recommendation:** 3

**Summary:**

The paper employs a multi-time stepping procedure in conjunction with a numerical integrator, physical operators, and neural networks to process downsampled data in both space and time. The proposed method applies time stepping on a finer temporal grid while utilizing the RK4 scheme for PDE integration. Supervision is provided through samples on coarser spatial-temporal grids, and the results demonstrate that this approach can yield accurate long-term predictions even when training data is limited.

**Claims And Evidence:**

The approach demonstrates efficiency in low training data regimes. Both quantitative and qualitative results underscore the effectiveness of the learning scheme, even when trained on a limited dataset (3–5 trajectories of 200–2000 timesteps), compared to the selected baseline methods.

**Essential References Not Discussed:**

To the best of my knowledge the related work is sufficient for the context of the key contributions.

**Experimental Designs Or Analyses:**

The experiments are designed to showcase the learning efficiency of the method. While they sufficiently demonstrate the method's efficiency relative to the baselines in a low training data regime, it would be beneficial to include tests with larger datasets to evaluate whether the observed improvements remain significant across varying data quantities.

**Methods And Evaluation Criteria:**

The method leverages physical priors, a numerical integration scheme, and learnable components to facilitate the efficient training of a PDE solver. The evaluation setting challenges current data-driven and physics-informed baselines by operating in a low training data regime. Given the high computational cost required to obtain reliable simulations, this approach is particularly noteworthy.

**Other Comments Or Suggestions:**

Typos:
- L241 (right colomn): "Sovling PDE".

**Other Strengths And Weaknesses:**

Strengths:
- The method's architecture is described in sufficient detail.

Weaknesses:
- The paper appears to test relatively few recent purely data-driven models as benchmarks. Could the authors provide an explanation or clarify the rationale for this choice? The claim on the amount of data should exclude all purely data-driven methods.
- Given the extent of physical prior information incorporated, it would be valuable to see the performance of the physics-informed counterparts of various architectures (e.g., FNO-PhyFNO, DeepONet-PhyDeepONet) across different cases. At a minimum, the authors should justify why certain baselines are included for some cases but not for others (e.g., PhyFNO is only evaluated for KdV).

**Questions For Authors:**

It appears that the "Correction Block" is an FNO without additional training objectives specifically aimed at achieving the correction. Could the authors clarify this design choice or provide qualitative results that illustrate how the state $\hat{u}_m^k$ is effectively corrected?

**Relation To Broader Scientific Literature:**

The key contributions of the paper lie in the continuous effort of the community on crossing different modeling principles and ideas from physics, numerical scheme and deep learning.

**Theoretical Claims:**

Not applicable.

---

> ### Author Rebuttal · Authors · 2025-04-01
>
> We appreciate your constructive comments. To enhance clarity, we have thoroughly proofread the manuscript and corrected all identified typographical errors. We believe these revisions will significantly improve the presentation. The updated version will be uploaded as soon as file submissions are enabled.
>
> ### **Weaknesses**
>
> >**W1. The benchmarks include few recent pure data-driven models.**
>
> **Reply:** Great remark! Our proposed MultiPDENet integrates PDE priors with multiscale time-stepping scheme, enabling efficient spatiotemporal simulations (e.g., turbulence) on coarse grids with limited data. While recent pure data-driven models (e.g., modern-UNet [1]) perform well in data-rich scenarios (e.g., requiring **128 trajectories**), their performance deteriorates significantly in small data settings.
>
> While we have selectd widely recognized data-driven baselines (e.g., FNO, modern-UNet, DeepONet), we have also considered hybrid physics-learning models (e.g., LI, TSM, PhyFNO, PeRCNN) for comparison. Following your suggestion, we added the recent model ONO [2] as an additional baseline. Its inferior performance under data scarcity (**see Table A**) further highlights the robustness of our framework.
>
> **Table A:** Comparison of MultiPDENet and baselines for NSE.
> |**Model**|**RMSE**(↓)|**MAE**(↓)|**MNAD**(↓)|**HCT**(s↑)|**Infer. cost**(s↓)|
> |-|-|-|-|-|-|
> |UNet|0.8224|0.5209|0.0627|3.9627|7|
> |FNO|1.0100|0.7319|0.0887|2.5749|5|
> |LI|NaN|NaN|NaN|3.5000|9|
> |TSM|NaN|NaN|NaN|3.7531|9|
> |DeepONet|2.1849|1.0227|0.1074|0.1126|**1**|
> |ONO|0.6613|0.4441|0.0535|4.2356|5|
> |MultiPDENet|**0.1379**|**0.0648**|**0.0077** |**8.3566**|26|
>
> >**W2. The selection of baselines for varying experimental cases.**
>
> **Reply:** Insightful comment! Our baseline selection carefully balances pure data-driven models (e.g., FNO, UNet, DeepONet) and physics-inspired counterparts (e.g., PhyFNO, LI, TSM, PeRCNN) to ensure fairness and relevance.
>
> To ensure consistent comparison, we added PhyFNO as a baseline for the Burgers and GS equations, with results shown in **Table B** below. So far, we have maintained a unified setting of baseline comparsion (FNO, UNet, DeepONet, PhyFNO, PeRCNN) for KdV, Burgers, and GS equations. In the case of NSE, we slightly altered the baseline models by considering two other well-recognized models specifically tailored for NSE (aka, LI and TSM). In all cases, we have demonstrated that MultiPDENet surpasses the baselines.
>
> We believe the current baseline setup ensures a **fair** and **consistent** comparison. Thanks for your great suggestion!
>
> **Table B:** Comparison of MultiPDENet and PhyFNO for Burgers and GS.
> |**Case**|**Model**|**RMSE**(↓)|**MAE**(↓)|**MNAD**(↓)|**HCT**(s↑)|
> |-|-|-|-|-|-|
> |Burgers|PhyFNO|0.0832|0.0749|0.0599|0.5546|
> |Burgers|MultiPDENet|**0.0057**|**0.0037**|**0.0031**|**1.4000**|
> |GS|PhyFNO|0.5721|0.3579|0.3520|510|
> |GS|MultiPDENet|**0.0573**|**0.0294**|**0.0298**|**1400**|
>
> ### **Suggestions**
> >**S1. Typos of "Sovling PDE".**
>
> **Reply:** We have thoroughly proofread the paper and corrected all typos in the revised version.
>
>
> ### **Questions**
>
> >**Q1. Design motivation for the Correction Block and the effectiveness of state correction.**
>
> **Reply:** Great remark! The Correction Block is designed to mitigate the information loss introduced by resolution reduction before computing the equivalent derivatives, enabling the model to adapt to the coarse grid. Instead of explicitly recovering the high-resolution solution, the Correction Block implicitly corrects the coarse-grid state by adjusting the scaling of the equivalent derivative term. During training, this block focuses on minimizing the overall PDE residual rather than directly reconstructing fine-grid details. The state $\hat{\bar{\mathbf{u}}}{_m^k}$, obtained via the Correction Block, represents a neural-corrected version of the coarse solution. However, this correction should not be interpreted as an explicit recovery of the fine-grid solution. Rather, it ensures that the equivalent derivative term computed via the Symmetric Filter is optimally adjusted to minimize the overall PDE residual. Our training objective is solely aimed at making the results more closely approximate the ground truth solution, as demonstrated in the ablation study (Model-E in **Table 3 on Page 8**). Therefore, there are no additional training objectives specifically designed for correction beyond this implicit adjustment mechanism.
>
> ***Refs:***
>
> [1] Gupta et al. Towards Multi-spatiotemporal-scale Generalized PDE Modeling. TMLR, 2023.
>
> [2] Xiao, et al. Improved Operator Learning by Orthogonal Attention. ICML, 2024.
>
> ***Remark:*** Once again, we sincerely appreciate your constructive comments. Looking forward to your feedback!

---

### Official Review · Reviewer_kUcx · 2025-03-09

**Overall Recommendation:** 5

**Summary:**

The paper introduces MultiPDENet, a PDE-embedded neural network with multiscale time stepping to accelerate flow simulations by integrating numerical methods with machine learning. It employs finite difference-based convolutional filters to approximate spatial derivatives on coarse grids, while a Physics Block with a 4th-order Runge-Kutta integrator preserves PDE structures for accurate predictions. To mitigate temporal error accumulation, a multiscale time integration approach is introduced, where a neural network corrects errors at a coarse time scale. Experiments on various PDEs, including the Navier-Stokes equations, demonstrate state-of-the-art accuracy and long-term stability with improved efficiency over traditional numerical methods and neural network baselines.

**Claims And Evidence:**

Yes, at least for the 2D NSE, it is pretty impressive to me.

**Essential References Not Discussed:**

There are missing some citations such as ICLR 2022 for black box methods.

**Experimental Designs Or Analyses:**

Yes, for all the dynamical systems.

**Methods And Evaluation Criteria:**

Yes, for fluid problem, the energy spectrum evaluation is good.

**Other Comments Or Suggestions:**

No, I do not have any.

**Other Strengths And Weaknesses:**

The paper needs to add a specific algorithm for training and a specific algorithm for testing.

**Questions For Authors:**

For NSE, is the white box performed on the coarse mesh?

Do we really need to satisfy the CFL condition?

For NSE, is the black box performed on the coarse mesh also?

**Relation To Broader Scientific Literature:**

I think the key contribution of the paper is demonstrating the black+white box method can work for NSE.

**Theoretical Claims:**

There is no such proofs for theoretical claims.

---

> ### Author Rebuttal · Authors · 2025-03-31
>
> Thanks for your constructive comments and suggestions! We have carefully addressed them, and the following responses have been incorporated into the revised paper.
>
> ### **Questions**
>
> >**Q1. Essential References Not Discussed.**
>
> **Re:** We appreciate your comment. We have noted that our initial submission omitted several important black-box methods in ICLR 2022, such as MP-PDE [1], GMR-Transformer-GMUS [2], and SiT [3]. We will include these references in the related work section in the revised manuscript.
>
> >**Q2. Is the white-box modules and black-blox moduls applied on the coarse mesh for NSE?**
>
> **Re:** Great remark! To accelerate the prediction of spatiotemporal dynamics on coarse grids, we integrated the numerical scheme with neural networks through a multiscale time-stepping strategy. Consequently, the entire network, encompassing both white-box and black-box modules, is designed to operate on a unfied coarse mesh.
>
> >**Q3. Does the model need to satisfy the CFL condition?**
>
> **Re:** Insightful question! According to the CFL condition, the maximum allowable time step is given by: $\delta t _{\max}=\mathrm{CFL} _{\max} \cdot \frac{\delta x}{\left| u \right| _{\max}}$, where the standard choice for numerical simulations is $\mathrm{CFL} _{\max}=0.5$. Based on this, we obtained $\delta t _{\max} = 0.007$ for the coarse grid with $\delta x = 2\pi/64$, which indeed means that our model satifies the CFL condition in our original test setting.
>
> However, to further investigate the extent to which MultiPDENet can go beyond the CFL condition, we conducted experiments with increased time intervals, namely, $\delta t_{\max} = 0.028$ (four times the required smallest time step). The resulting model is called MultiPDENet-L. With this setup, we achieved an extra 4$\times$ speedup (e.g., inference time of 6 sec vs. the original 26 sec) while maintaining a similar model accuracy, aligning with the speed of other baseline models as shown in **Table A**. Note that MultiPDENet-L was trained based on a rollout strategy over 8 macro-steps.
>
> In summary, MultiPDENet does **not** need to adhere to the CFL condition, demonstrating its capability to operate effectively beyond conventional stability constraints.
>
> **Table A:** Comparison of MultiPDENet and baselines for NSE
> |Model|RMSE(↓)|MAE(↓)|MNAD(↓)|HCT(s↑)|Infer. cost(s↓)|
> |-|-|-|-|-|-|
> |UNet|0.82|0.52|0.06|3.96|7|
> |FNO|1.01|0.73|0.09|2.57|5|
> |LI|NaN|NaN|NaN|3.50|9|
> |TSM|NaN|NaN|NaN|3.75|9|
> |DeepONet|2.18|1.02|0.11|0.11|**1**|
> |MultiPDENet|**0.13**|**0.06**|**0.01**|**8.36**|26|
> |MultiPDENet-L |0.37  |0.18  | 0.02 | 7.42 |6|
>
> ***Refs:***
>
> [1] Brandstetter et al. Message passing neural PDE solvers. ICLR, 2022.
>
> [2] Han et al. Predicting Physics in Mesh-reduced Space with Temporal Attention. ICLR, 2022.
>
> [3] Shao et al. SiT: Simulation transformer for particle-based physics simulation. ICLR, 2022.
>
> ***Remark:*** Once again, we sincerely appreciate your constructive comments. Please feel free to let us know if you have any further questions. Looking forward to your feedback!

---

> > ### Comment · Reviewer_kUcx · 2025-04-08
> >
> > I appreciate the authors’ effort for addressing my comments. However, I still have some questions. Firstly, I kindly ask an algorithm of training and another algorithm of testing (evaluation) for the fluid problems. Secondly, I want a table to list the solvers used for all cases in the paper, and also show if the solvers are open-sourced, I believe this will benefit the community. Thirdly, for Poisson block used in the NS, does the Poisson block iteratively solve the pressure equation? Or it is a direct solving. Fourthly, since A100 is used, how much GPU memory is needed to train and evaluate the model for the NS case, I want an exact number. I am looking forward to the authors reply.

---

> > > ### Author Response · Authors · 2025-04-08
> > >
> > > We appreciate your additional comments and trust that the following responses will address the concerns raised. These responses will be incorporated into our revised paper.
> > >
> > >
> > > ### **Questions**
> > >
> > > >**Q1. The training and testing algorithm for the fluid problems.**
> > >
> > > **Re:** Great remark! In fluid dynamics problems, the autoregressive roll-out training method is most commonly adopted for long-term prediction tasks [1]. A key advantage of this approach lies in its flexible adjustment of the roll-out window size tailored to specific scenarios. For example, in NSE problems, a 32-step roll-out window is often empirically chosen as a balance between computational efficiency and numerical stability [2, 3]. When model stability is ensured, training strategies with single-step predictions (i.e., larger time steps) can be employed to reduce computational overhead [4]. In this work, we introduce a micro-step correction mechanism to stabilize the network’s performance over macro-step predictions, enabling reliable single-step roll-out training while maintaining accuracy.
> > >
> > > For testing, the standard algorithm involves comparing predicted trajectories against ground truth using statistical metrics (e.g., RMSE, HCT, MNAD) and physics-consistency metrics (e.g., Energy spectrum). The former metrics quantify numerical accuracy, and the latter evaluate adherence to fundamental fluid mechanics principles. This combination of evaluations ensures that predictions are not only numerically precise but also physically interpretable and robust.
> > >
> > > >**Q2. Solver used in the paper.**
> > >
> > > **Re:** We appreciate your comment! All numerical solvers employed in this study are summarized in **Table A**, and their implementations will be open-sourced alongside the reproduction toolkit. This code release aims to facilitate both the replication of our experiments and extended research endeavors by the community.
> > >
> > > **Table A:** Numerical Solver for generating datasets
> > > |Cases|Numerical method|Spatial grid|Temporal steps|Open-sourced|
> > > |-|-|-|-|-|
> > > |KdV|Spectral|256|10000|yes|
> > > |Burgers|FD|100$^2$|2000|yes|
> > > |GS|FD|128$^2$|4000|yes|
> > > |NSE|FV|2048$^2$|153600|yes|
> > >
> > >
> > > >**Q3. Does the Poisson block iteratively solve the pressure equation?**
> > >
> > > **Re:** Excellent comment! The Poisson block, which is critical for solving the pressure term, relies on the Poisson solver as its core component. This solver leverages a numerical methodology grounded in frequency-domain. The basic idea is to convert the original problem into the frequency domain, where the spatial differential operation becomes a multiplication operation via Fourier transform. The Poisson equation is then solved in the frequency domain, and the inverse Fourier transform is then used to obtain the solution in the original spatial domain. From the Navier-Stokes equations, we derive the relation $\Delta p = 2 \left(u_x v_y - u_y v_x\right)$ (the subscripts indicate the spatial derivatives along $x$ or $y$ directions). By directly feeding the right-hand side of this expression into the Poisson solver, we can compute the pressure term without requiring an iterative process.
> > >
> > >
> > > >**Q4. GPU usage for training and testing on Navier-Stokes Cases.**
> > >
> > > **Re:** Thanks for this comment! When training on the NSE, we use a batch size of 90, which results in a GPU memory usage of 74.79 GiB. During inference, the memory consumption is significantly lower, at 12.38 GiB. Notably, our model can also be trained and deployed on a single RTX 4090 GPU by adjusting the batch size to 20, which requires only 19.50 GiB of GPU memory.
> > >
> > >
> > > ***Refs:***
> > >
> > > [1] Rao, et al. Encoding physics to learn reaction–diffusion processes. NMI, 2023.
> > >
> > > [2] Kochkov, et al. Machine learning–accelerated computational fluid dynamics. PNAS, 2021.
> > >
> > > [3] Sun, et al. A Neural PDE Solver with Temporal Stencil Modeling. ICLR, 2023.
> > >
> > > [4] K. Gupta, et al. Towards Multi-spatiotemporal-scale Generalized PDE Modeling. TMLR, 2023.
> > >
> > > ***Remark:*** Thank you very much for your valuable feedback. Please let us know if you have other questions!

---

### Official Review · Reviewer_HqxA · 2025-03-14

**Overall Recommendation:** 3

**Summary:**

The authors proposed a framework for data-driven fluid flow simulation on uniform grid. Trying to soft-embed partial differential equations in data-driven flow simulations, the authors design a convolutional filter based on the constraints of the central difference discretization of the first and second order derivatives. To be able to achieve accurate large time step rollout predictions in the learned model, the authors generate each time step prediction by first iterate through a series of pseudo time steps through a "physics block", and then apply a corrector at the end to generate the corrected predictions at the next time step. Test results show that the proposed method outperforms existing benchmarks.

**Claims And Evidence:**

Yes

**Essential References Not Discussed:**

No

**Experimental Designs Or Analyses:**

All experimental setups have been reviewed. Please see "questions for authors" for additional concerns and questions I have.

**Methods And Evaluation Criteria:**

Yes. The method itself leads to a series of limitations though, see "other strength and weakness" for details.

**Other Comments Or Suggestions:**

N/A

**Other Strengths And Weaknesses:**

1. It should be noted that the proposed method not only has to be applied to structured grid (as is with most networks that utilize convolutions), but also has to be applied on uniform grid due to the limitation of the filters used. This is a significant limitation of this work which is likely not possible to resolve.
2. The authors did not discuss the extension to 3D cases.
3. The proposed framework will likely to suffer in cases with irregular geometries. The boundaries will be concerning when objects of non-cubic shapes are involved, since the authors only provide strategies to handle periodic boundaries of simple shape.

**Questions For Authors:**

1. Table S7 & S8, am I understanding it correctly that the model is only faster than Direct Numerical Simulation (which is well-known to be computationally expensive) by 5-7 times?
2. If that is the case then I am deeply concerned about the inference speed of the proposed framework, since typical architectures in the domain are usually reported to be orders of magnitude faster than unsteady RANS or LES (both are cheaper than DNS due to less requirement on grid density). Please report the inference speed of the proposed network versus other benchmark cases.
3. Continued from 2, please report the performance of different models when their inference speed is about the same, by adjusting the size of the models. It should be noted that such comparison should be performed by shrinking the size of the proposed network rather than increasing the size of the benchmark networks, as the benchmark networks should stay at around the recommended sizes in their respective papers.

If the authors can prove with sufficient evidence that the proposed framework still outperforms benchmark cases when the networks are adjusted to run at the same inference time per step, then I am happy to raise the score to 3.

**Relation To Broader Scientific Literature:**

The paper is related to the data-driven modeling of (fluid) flow. The authors have provided enough reviews of the related literature in the introduction.

**Theoretical Claims:**

N/A

---

> ### Author Rebuttal · Authors · 2025-03-31
>
> Thanks for your constructive comments! We have addressed them thoroughly and added new figures/tables (see the **rebuttal.pdf** via https://anonymous.4open.science/r/Rebuttal-5D8B/rebuttal.pdf). These results will be added to the revised paper.
>
>
> ### **Weaknesses**
>
> >**W1. Uniform grids.**
>
> **Re:** Thanks for your **thoughtful comments** on irregular geometries and complex boundary conditions (BCs). Our study focuses on fluid simulations on regular grids with periodic BCs, a common constraint shared with methods like FNO, LI, and TSM. This indicates that the limitation is not specific to MultiPDENet but represents a general challenge in the field. Similar to Geo-FNO [1], a geometry-aware netwok (e.g., a geometry encoder-decoder) could be established on top of MultiPDENet (latent structured geometry learner) to handle general geometries. Thank you for pointing out the important direction for our future work!
>
> To demonstrate our model's generalization to complex BCs, we conducted experiments on Burgers equation with Dirichlet and Neumann BCs, while keeping other settings consistent with the original data generation setup. We tested our previously trained model for inference on 10 trajectories with complex BCs through BC encoding (**Table R1** in **rebuttal.pdf**). **Figure R1** in **rebuttal.pdf** shows predicted snapshots, confirming the model's generalizability over complex BCs.
>
> >**W2. Lacking 3D cases.**
>
> **Re:** Great remark! We followed your comment and tested our model on the 3D Gray-Scott (GS) equation. We generated 5 datasets (1 for training and 4 for testing).
>
> Snapshots of trajectory evolution from 0 to 600 s for MultiPDENet and baselines are shown in **Figure R2(a)** in **rebuttal.pdf**. **Figure R2(b)** in **rebuttal.pdf** shows MultiPDENet maintains a Pearson correlation coefficient $>$0.8 throughout the evolution. The error distribution in **Figure R2\(c)** in **rebuttal.pdf** highlights our model’s superior performance. **Table R2** in **rebuttal.pdf** summarizes our model’s performance suggesting strong potential to solve 3D problems.
>
> ### **Questions**
>
> >**Q1. Model inference speed.**
>
> **Re:** Thanks for raising this important point! It's crucial to **clarify** that the reported orders of magnitude speedups for popular models like FNO, LI and DeepONet (10$^3\times$, 80$\times$, 24$\times$) are often **not** evaluated against numerical methods with comparable accuracy. McGreivy et al. [2] revealed this by reimplementing these models and comparing them under consistent precision, resulting in only 7$\times$ speedup for FNO, while LI and DeepONet exhibited slower performance. Their evaluation effectively highlighted the prior discrepancies in speedup reporting.
>
> Consistent with [2], we demonstrate a 5–7$\times$ speedup under the condition of sonsistent accuracy, althouh implementation in JAX may further yield extra speedup gains (e.g., $>5\times$) [3]. We must admit that the rollout strategy used in our model inherently limits its speed, while reducing the network size provides only marginal gains. Nonetheless, MultiPDENet generalizes well across diverse initial conditions, varying $Re$, external forces, and larger domains.
>
> >**Q2. Does MultiPDENet still outperform other models at the same inference speed?**
>
> **Re:** Great question! As mentioned above, solely reducing network parameters does not substantially accelerate the inference due to the employed rollout strategy. However, leveraging a larger time step $\delta t$ proves effective. MultiPDENet's multiscale time-stepping design allows it to circumvent the CFL condition, ensuring accuracy and stability even with increased $\delta t$. Specifically, with $4\delta t$ (MultiPDENet-L), we achieved an extra 4$\times$ speedup (e.g., inference time of 6s) while maintaining a similar model accuracy, aligning with the speed of other models as shown in **Table A**. Note that MultiPDENet-L was trained based on a rollout strategy over 8 macro-steps.
>
> These results demonstrate our model’s multiscale time stepping scheme consistently outperforms all baselines across all metrics, even at similar inference speeds. Hope this clarifies your concern.
>
> **Table A:** Comparison of MultiPDENet and baselines for NSE
> |Model|RMSE(↓)|MNAD(↓)|HCT(s↑)|Infer. cost(s↓)|
> |-|-|-|-|-|
> |UNet|0.82|0.06|3.96|7|
> |FNO|1.01|0.09|2.57|5|
> |LI|NaN|NaN|3.50|9|
> |TSM|NaN|NaN|3.75|9|
> |DeepONet|2.18|1.02|0.11|**1**|
> |MultiPDENet|**0.13**|**0.01**|**8.36**|26|
> |MultiPDENet-L|0.37|0.02|7.43|6|
>
> ***Refs:***
>
> [1] Li et al. Fourier neural operator with learned deformations for pdes on general geometries. JLMR, 2023
>
> [2] McGreivy et al. Weak baselines and reporting biases lead to overoptimism in machine learning for fluid-related partial differential equations. Nature Machine Intelligence, 2024.
>
> [3] Takamoto et al. PDEbench: An extensive benchmark for scientific machine learning. NeurIPS, 2022.
>
> **Remark:** Please let us know if you have other questions. Looking forward to your feedback!

---

> > ### Comment · Reviewer_HqxA · 2025-04-04
> >
> > Thank you for the response. I think my concerns are largely addressed. I am raising the score to 3.

---

> > > ### Author Response · Authors · 2025-04-04
> > >
> > > Thank you for your positive feedback and for increasing the score. We will include the additional experiments and text in the revised paper.

---

### Decision · Program_Chairs · 2025-05-01

**Decision:**

Accept (poster)

**Comment:**

This paper studies a hybrid scheme to simulate Navier-Stokes models. The marching still follows traditional multi-step methods (RK4 is used in the paper), while the overall forcing uses a hybrid scheme (pressure is solved using an FFT Poisson solver, the convection part is taken care by NNs). All reviewers give positive feedbacks toward accept. However, one review was too short so I read the paper myself. Below are some comments:

- Equation (1), if $\mathbf{u}$ is a vector, I would not use $\mathbf{u}^2$ notation, and this confuses even more as later on $\mathbf{u}^k$ is used to denote the snapshots at the sub-timesteps.
- Sometimes, the writing uses M${}_i$ NN, while some other times, it is MiNN.
- Page 8: "confirming p-u decoupling necessity in the NSE." It should be "coupling". Also I suggest adding another ablation on the different placement of the Poisson block (e.g., after MiNN).

I also suggest the authors add the following references:
- Another post-processing scheme to correct surrogate models: Lippe et al. *PDE-Refiner: Achieving Accurate Long Rollouts with Neural PDE Solvers*.
- Parametrizing filters with symmetric (or skew-) contraint: Luan et al. *Gabor Convolutional Networks*.
- This paper studies the learning of particle-based time steppers (with a traditional integrator), similar to this submission: Li & Farimani: *Graph neural network-accelerated Lagrangian fluid simulation*.
- Using Poisson solver explicitly in substep methods, instead of solving a mixed formulation of divergence-free equation: Guermond, Minev, Shen. Comput. Methods Appl. Mech. Engrg. 195 (2006) 6011–6045.